# SHARPER GENERALIZATION BOUNDS FOR LEARNING WITH GRADIENT-DOMINATED OBJECTIVE FUNCTIONS

**Yunwen Lei**[1,2]   **Yiming Ying**[3*]

[1]School of Computer Science, University of Birmingham, Birmingham B15 2TT, United Kingdom
[2]Department of Computer Science, TU Kaiserslautern, Kaiserslautern 67653, Germany
[3]Department of Mathematics and Statistics, State University of New York at Albany, USA
`y.lei@bham.ac.uk`  `yying@albany.edu`

## ABSTRACT

Stochastic optimization has become the workhorse behind many successful machine learning applications, which motivates a lot of theoretical analysis to understand its empirical behavior. As a comparison, there is far less work to study the generalization behavior especially in a non-convex learning setting. In this paper, we study the generalization behavior of stochastic optimization by leveraging the algorithmic stability for learning with $\beta$-gradient-dominated objective functions. We develop generalization bounds of the order $O(1/(n\beta))$ plus the convergence rate of the optimization algorithm, where $n$ is the sample size. Our stability analysis significantly improves the existing non-convex analysis by removing the bounded gradient assumption and implying better generalization bounds. We achieve this improvement by exploiting the smoothness of loss functions instead of the Lipschitz condition in Charles & Papailiopoulos (2018). We apply our general results to various stochastic optimization algorithms, which show clearly how the variance-reduction techniques improve not only training but also generalization. Furthermore, our discussion explains how interpolation helps generalization for highly expressive models.

## 1  INTRODUCTION

Stochastic optimization has found tremendous applications in training highly expressive machine learning models including deep neural networks (DNNs) (Bottou et al., 2018), which are ubiquitous in modern learning architectures (LeCun et al., 2015). Oftentimes, the models trained in this way have not only very small training errors or even interpolate the training examples, but also surprisingly generalize well to testing examples (Zhang et al., 2017). While the low training error can be well explained by the over-parametrization of models and the efficiency of the optimization algorithm in identifying a local minimizer (Bassily et al., 2018; Vaswani et al., 2019; Ma et al., 2018), it is still unclear how the highly expressive models also achieve a low testing error (Ma et al., 2018). With the recent theoretical and empirical study, it is believed that a joint consideration of the interaction among the optimization algorithm, learning models and training examples is necessary to understand the generalization behavior (Neyshabur et al., 2017; Hardt et al., 2016; Lin et al., 2016).

The generalization error for stochastic optimization typically consists of an optimization error and an estimation error (see e.g. Bousquet & Bottou (2008)). Optimization errors arise from the suboptimality of the output of the chosen optimization algorithms, while estimation errors refer to the discrepancy between the testing error and training error at the output model. There is a large amount of literature on studying the optimization error (convergence) of stochastic optimization algorithms (Bottou et al., 2018; Orabona, 2014; Karimi et al., 2016; Ying & Zhou, 2017; Liu et al., 2018). In particular, the power of interpolation is clearly justified in boosting the convergence rate of stochastic gradient descent (SGD) (Bassily et al., 2018; Vaswani et al., 2019; Ma et al., 2018). In contrast, there is far less work on studying estimation errors of optimization algorithms. In a seminal paper (Hardt et al., 2016), the fundamental concept of algorithmic stability was used to study the generalization behavior of SGD, which was further improved and extended in Charles & Papailiopoulos (2018); Zhou et al. (2018b); Yuan et al. (2019); Kuzborskij & Lampert (2018).

---

*Corresponding author: Yiming Ying

However, these results are still not quite satisfactory in the following three aspects. Firstly, the existing stability bounds in non-convex learning require very small step sizes (Hardt et al., 2016) and yield suboptimal generalization bounds (Yuan et al., 2019; Charles & Papailiopoulos, 2018; Zhou et al., 2018b). Secondly, majority of the existing work has focused on functions with a uniform Lipschitz constant which can be very large in practical models if not infinite (Bousquet & Elisseeff, 2002; Hardt et al., 2016; Charles & Papailiopoulos, 2018; Kuzborskij & Lampert, 2018), e.g., DNNs. Thirdly, the existing stability analysis fails to explain how the highly expressive models still generalize in an interpolation setting, which is observed for overparameterized DNNs (Arora et al., 2019; Brutzkus et al., 2017; Bassily et al., 2018; Belkin et al., 2019).

In this paper, we make attempts to address the above three issues using novel stability analysis approaches. Our main contributions are summarized as follows.

1. We develop general stability and generalization bounds for *any* learning algorithm to optimize (non-convex) $\beta$-gradient-dominated objectives. Specifically, we show that the excess generalization error is bounded by $O(1/(n\beta))$ plus the convergence rate of the algorithm, where $n$ is the sample size. This general theorem implies that overfitting will never happen in this case, and generalization would always improve as we increase the training accuracy, which is due to an implicit regularization effect of gradient dominance condition. In particular, we show that interpolation actually improves generalization for highly expressive models. In contrast to the existing discussions based on either hypothesis stability or uniform stability which imply at best a bound of $O(1/\sqrt{n\beta})$, the main idea is to consider a weaker on-average stability measure which allows us to replace the uniform Lipschitz constant in Hardt et al. (2016); Kuzborskij & Lampert (2018); Charles & Papailiopoulos (2018) with the training error of the best model.

2. We apply our general results to various stochastic optimization algorithms, and highlight the advantage over existing generalization analysis. For example, we derive an exponential convergence of testing errors for SGD in an interpolation setting, which complements the exponential convergence of optimization errors (Bassily et al., 2018; Vaswani et al., 2019; Ma et al., 2018) and extends the existing results (Pillaud-Vivien et al., 2018; Nitanda & Suzuki, 2019) from a strongly-convex setting to a non-convex setting. In particular, we show that stochastic variance-reduced optimization outperforms SGD by achieving a significantly faster convergence of testing errors, while this advantage is only shown in terms of optimization errors in the literature (Reddi et al., 2016; Lei et al., 2017; Nguyen et al., 2017; Zhou et al., 2018a; Wang et al., 2019).

## 2 RELATED WORK

**Algorithmic Stability**. We first review the related work on stability and generalization. Algorithmic stability is a fundamental concept in statistical learning theory (Bousquet & Elisseeff, 2002; Elisseeff et al., 2005), which has a deep connection with learnability (Shalev-Shwartz et al., 2010; Rakhlin et al., 2005). The important uniform stability was introduced in Bousquet & Elisseeff (2002), where the authors showed that empirical risk minimization (ERM) enjoys the uniform stability if the objective function is strongly convex. This concept was extended to study randomized algorithms such as bagging and bootstrap (Elisseeff et al., 2005). An interesting trade-off between uniform stability and convergence was developed for iterative optimization algorithms, which was then used to study convergence lower bounds of different algorithms (Chen et al., 2018). While generalization bounds based on stability are often stated in expectation, uniform stability was recently shown to guarantee almost optimal high-probability bounds based on elegant concentration inequalities for weakly-dependent random variables (Maurer, 2017; Feldman & Vondrak, 2019; Bousquet et al., 2020). Other than the standard classification and regression setting, uniform stability was very successfully to study transfer learning (Kuzborskij & Lampert, 2018), PAC-Bayesian bounds (London, 2017), privacy learning (Bassily et al., 2019) and pairwise learning (Lei et al., 2020b). Some other stability measures include the uniform argument stability (Liu et al., 2017), hypothesis stability (Bousquet & Elisseeff, 2002), hypothesis set stability (Foster et al., 2019) and on-average stability (Shalev-Shwartz et al., 2010). An advantage of on-average stability is that it is weaker than the uniform stability and can imply better generalization by exploiting either the strong convexity of the objective function (Shalev-Shwartz & Ben-David, 2014, Corollary 13.7) or the more relaxed exp-concavity of loss functions (Koren & Levy, 2015; Gonen & Shalev-Shwartz, 2017). Since gradient-dominance condition is another relaxed extension of strong convexity, we use on-average stability to study generalization bounds.

**Generalization analysis**. We now review related work on generalization analysis for stochastic optimization. In a seminal paper (Hardt et al., 2016), the authors used the nonexpansiveness of gradient mapping to develop uniform stability bounds for SGD to optimize convex, strongly convex and even non-convex objective functions. This inspired some interesting work on stochastic optimization. An interesting data-dependent stability bound was developed for SGD, a nice property of which is that it shows how the initialization would affect generalization (Kuzborskij & Lampert, 2018). These stability bounds were integrated into a PAC-Bayesian analysis of SGD, yielding generalization bounds for arbitrary posterior distributions (London, 2017). Almost optimal generalization bounds were developed for differentially private stochastic convex optimization (Bassily et al., 2019). The on-average variance of stochastic gradients was used to refine the generalization analysis of SGD (Hardt et al., 2016) in non-convex optimization (Zhou et al., 2018b). The uniform stability was also studied for SGD implemented in a stagewise manner (Yuan et al., 2019) and stochastic gradient Langevin dynamics in a non-convex setting (Li et al., 2020; Mou et al., 2018). Very recently, the discussions in Hardt et al. (2016) were extended to tackle non-smooth (Lei & Ying, 2020; Bassily et al., 2020) and non-Lipscthiz functions (Lei & Ying, 2020). The most related work is Charles & Papailiopoulos (2018), where some general hypothesis stability bounds were developed for learning algorithms that converge to optima. A very interesting point is that their bounds depend only on the convergence of the algorithm to a global minimum and the geometry of loss functions around the global minimum. However, their discussion imply at best the slow generalization bounds $O(1/\sqrt{n\beta})$ for $\beta$-gradient-dominated objective functions, and can not explain the benefit of low optimization errors in helping generalization. The underlying reason is that they used the pointwise hypothesis stability and did not consider the smoothness of loss functions. We aim to improve these results by leveraging the weaker on-average stability and smoothness of loss functions.

Other than the stability approach, there is interesting generalization analysis of SGD based on either a uniform convergence approach (Lin et al., 2016), an integral operator approach (Lin & Rosasco, 2017; Ying & Pontil, 2008; Dieuleveut & Bach, 2016; Dieuleveut et al., 2017; Mücke et al., 2019) or an information-theoretic approach (Xu & Raginsky, 2017; Negrea et al., 2019; Bu et al., 2020).

## 3   MAIN RESULTS

Let $\rho$ be a probability measure defined on a sample space $\mathcal{Z} = \mathcal{X} \times \mathcal{Y}$ with $\mathcal{X} \subseteq \mathbb{R}^d$ and $\mathcal{Y} \subseteq \mathbb{R}$, from which a training dataset $S = \{z_1, \ldots, z_n\}$ is drawn independently and identically. The aim is to find a good model $\mathbf{w}$ from a model parameter space $\mathcal{W}$ based on the training dataset $S$. The performance of a prescribed model $\mathbf{w}$ on a single example $z$ can be measured by a nonnegative loss function $f(\mathbf{w}; z)$, where $f : \mathcal{W} \times \mathcal{Z} \mapsto \mathbb{R}_+$. In machine learning we often apply an (randomized) algorithm $A : \cup_n \mathcal{Z}^n \mapsto \mathcal{W}$ to $S$ to produce an output model $A(S) \in \mathcal{W}$. Oftentimes, the constructed model $\mathbf{w}$ would have a small *empirical risk* $F_S(\mathbf{w}) = \frac{1}{n} \sum_{i=1}^n f(\mathbf{w}; z_i)$. However, we are mostly interested in the generalization performance of a model $\mathbf{w}$ on testing examples measured by the *population (true) risk* $F(\mathbf{w}) = \mathbb{E}_z[f(\mathbf{w}; z)]$, where $\mathbb{E}_z$ denotes the expectation with respect to (w.r.t.) $z$. The gap $\mathbb{E}_{S,A}[F(A(S)) - F_S(A(S))]$ between the population risk and empirical risk is called the *estimation error*, which is due to the approximation of $\rho$ by sampling. Here $\mathbb{E}_A$ denotes the expectation w.r.t. the randomness of the algorithm $A$. For example, if $A$ is SGD, then $\mathbb{E}_A$ denotes the expectation w.r.t. the random indices of training examples selected for the gradient computation. A powerful tool to study the estimation error is the algorithmic stability (Bousquet & Elisseeff, 2002; Elisseeff et al., 2005; Shalev-Shwartz et al., 2010; Hardt et al., 2016), which measures the sensitivity of the algorithm's output w.r.t. the perturbation of a training dataset. Below we give formal definitions of stability measures, whose connection to generalization is established in Theorem A.1.

**Definition 1** (Uniform Stability). A randomized algorithm $A$ has uniform stability $\epsilon$ if for all datasets $S, \widetilde{S} \in \mathcal{Z}^n$ that differ by at most one example, we have $\sup_z \mathbb{E}_A[f(A(S); z) - f(A(\widetilde{S}); z)] \leq \epsilon$.

The following on-average stability is similar to the average-RO stability in Shalev-Shwartz et al. (2010). The difference is we do not use an absolute value. For $m \in \mathbb{N}$, we denote $[m] = \{1, \ldots, m\}$.

**Definition 2** (On-average Stability). Let $S = \{z_1, \ldots, z_n\}$ and $\widetilde{S} = \{\tilde{z}_1, \ldots, \tilde{z}_n\}$ be drawn independently from $\rho$. For each $i \in [n]$, denote $S^{(i)} = \{z_1, \ldots, z_{i-1}, \tilde{z}_i, z_{i+1}, \ldots, z_n\}$. We say an algorithm $A$ has on-average stability $\epsilon$ if $\frac{1}{n} \sum_{i=1}^n \mathbb{E}_{S,\widetilde{S},A}[f(A(S^{(i)}); z_i) - f(A(S); z_i)] \leq \epsilon$.

In this paper, we are interested in the *excess generalization error* $F(A(S)) - F(\mathbf{w}^*)$, where $\mathbf{w}^* \in \arg\min_{\mathbf{w} \in \mathcal{W}} F(\mathbf{w})$ is the best model with the least testing error (population risk). For this purpose, we introduce some basic assumptions. A basic assumption in non-convex learning is the smoothness of loss functions (Ghadimi & Lan, 2013; Karimi et al., 2016), meaning the gradients are Lipschitz continuous. Let $\|\cdot\|_2$ denote the Euclidean norm and $\nabla$ denote the gradient operator.

**Assumption 1** (Smoothness Assumption). We assume for all $z \in \mathcal{Z}$, the differentiable function $\mathbf{w} \mapsto f(\mathbf{w}; z)$ is $L$-smooth, i.e., $\|\nabla f(\mathbf{w}; z) - \nabla f(\mathbf{w}'; z)\|_2 \leq L\|\mathbf{w} - \mathbf{w}'\|_2$ for all $\mathbf{w}, \mathbf{w}' \in \mathcal{W}$.

Another assumption is the Polyak-Lojasiewicz (PL) condition on the objective function, which is common in non-convex optimization (Zhou et al., 2018b; Reddi et al., 2016; Karimi et al., 2016; Wang et al., 2019; Lei et al., 2017), and was shown to hold true for deep (linear) and shallow neural networks (Hardt & Ma, 2016; Charles & Papailiopoulos, 2018; Li & Yuan, 2017).

**Assumption 2** (Polyak-Lojasiewicz Condition). Denote $\hat{F}_S = \inf_{\mathbf{w}' \in \mathcal{W}} F_S(\mathbf{w}')$. We assume $F_S$ satisfies PL or gradient-dominated condition (in expectation) with parameter $\beta > 0$, i.e.,

$$\mathbb{E}_S\big[F_S(\mathbf{w}) - \hat{F}_S\big] \leq \frac{1}{2\beta}\mathbb{E}_S\big[\|\nabla F_S(\mathbf{w})\|_2^2\big], \quad \forall \mathbf{w} \in \mathcal{W}. \tag{3.1}$$

It is worthy of mentioning that our results in this section continue to hold if the global PL condition is relaxed to a local PL condition, i.e., (3.1) holds for $\mathbf{w}$ in a neighborhood of the minimizer of $F_S$.

The existing stability analysis often imposes a bounded gradient assumption below (Bousquet & Elisseeff, 2002; Hardt et al., 2016; Charles & Papailiopoulos, 2018; Yuan et al., 2019; Kuzborskij & Lampert, 2018). Indeed, the resulting stability bounds depend on the uniform Lipschitz constant $G$ (see eq. (3.4)), which can be prohibitively large in practical models, e.g., DNNs, or even infinite, e.g. least squares regression in an unbounded domain.

**Assumption 3** (Bounded Gradient Assumption). We assume $\|\nabla f(\mathbf{w}; z)\|_2 \leq G$ for all $\mathbf{w} \in \mathcal{W}$, $z \in \mathcal{Z}$ and a constant $G > 0$.

Our main result to be proved in Appendix B removes Assumption 3 and replaces the uniform Lipschitz constant $G$ by the minimal empirical risk $\hat{F}_S$, which is significantly smaller than the Lipschitz constant. Note the assumption $L \leq n\beta/4$ is mild, and the previous generalization bounds become vacuous as $O(1)$ (Yuan et al., 2019; Charles & Papailiopoulos, 2018) if this assumption is violated.

**Theorem 1** (Main Theorem). *Let Assumptions 1, 2 hold and $\mathbf{w}_S = A(S)$. If $L \leq n\beta/4$, then*

$$\mathbb{E}\big[F(\mathbf{w}_S) - \hat{F}_S\big] \leq \frac{16L\mathbb{E}[\hat{F}_S]}{n\beta} + \frac{L\mathbb{E}[F_S(\mathbf{w}_S) - \hat{F}_S]}{2\beta}. \tag{3.2}$$

An important implication is as follows. Since $\mathbb{E}\big[\hat{F}_S\big] \leq \mathbb{E}\big[F_S(\mathbf{w}^*)\big] = F(\mathbf{w}^*)$ and $\hat{F}_S \leq F_S(\mathbf{w}_S)$, Eq. (3.2) implies an upper bound on the excess generalization error $\mathbb{E}[F(\mathbf{w}_S)] - F(\mathbf{w}^*)$ and

$$\mathbb{E}\big[F(\mathbf{w}_S) - F_S(\mathbf{w}_S)\big] = O\Big(\frac{1}{n\beta} + \frac{\mathbb{E}\big[F_S(\mathbf{w}_S) - \hat{F}_S\big]}{\beta}\Big). \tag{3.3}$$

The above two terms can be explained as follows. The term $O(1/(n\beta))$ reflects the intrinsic complexity of the problem, while $\mathbb{E}\big[F_S(\mathbf{w}_S) - \hat{F}_S\big]$ is called the *optimization error*. An interesting observation is that the overfitting phenomenon would never happen for learning under the PL condition (analogous to learning with strongly convex objectives where the global minimizer generalizes well (Bousquet & Elisseeff, 2002)). Indeed, if the optimization algorithm finds more and more accurate solutions, it achieves the limiting generalization bound $O(1/(n\beta))$. This shows an important message that optimization can be beneficial to generalization. This seemingly counterintuitive phenomenon is due to the *implicit regularization* enforced by the PL condition (analogous to the strong convexity condition). Another notable property is that Theorem 1 applies to any algorithm. We can plug any known optimization error bounds into it to immediately get generalization bounds.

**Remark 1.** We show that our result significantly improves the existing stability analysis. The work (Charles & Papailiopoulos, 2018) showed the pointwise hypothesis stability is controlled by $\frac{2G^2}{n\beta} + 2\sqrt{2}G\sqrt{\mathbb{E}[F_S(\mathbf{w}_S) - \hat{F}_S]/\beta}$, which together with the connection between stability and generalization (cf. (A.1)), implies with probability $1 - \delta$ that

$$F(\mathbf{w}_S) \leq F_S(\mathbf{w}_S) + \Big(\frac{M^2}{n\delta} + \frac{24MG^2}{n\beta\delta} + \frac{24MG\sqrt{2\mathbb{E}[F_S(\mathbf{w}_S) - \hat{F}_S]}}{\sqrt{\beta}\delta}\Big)^{\frac{1}{2}}. \tag{3.4}$$

The above bound requires the bounded gradient assumption $\|\nabla f(\mathbf{w}; z)\|_2 \leq G$ and the bounded loss assumption $0 \leq f(\mathbf{w}; z) \leq M$ for all $\mathbf{w} \in \mathcal{W}$ and $z \in \mathcal{Z}$, which are successfully removed in our generalization analysis. Furthermore, our generalization bound significantly improves (3.4). Indeed, assume $\mathbb{E}[F_S(\mathbf{w}_S) - \hat{F}_S] \leq \epsilon^2 \beta$ for some $\epsilon > 0$, then (3.3) implies

$$\mathbb{E}\big[F(\mathbf{w}_S)\big] = \mathbb{E}\big[F_S(\mathbf{w}_S)\big] + O\Big(\frac{1}{n\beta} + \epsilon^2\Big), \tag{3.5}$$

while (3.4) becomes $F(\mathbf{w}_S) = F_S(\mathbf{w}_S) + O\Big(\frac{1}{\sqrt{n\beta}} + \sqrt{\epsilon}\Big)$. To achieve the generalization guarantee $O(1/\sqrt{n\beta})$, the above bound requires the optimization accuracy $\epsilon = O(1/(n\beta))$, while our bound (3.5) only requires the accuracy $\epsilon = O(1/\sqrt{n\beta})$ but gets the significantly better generalization bound $1/(n\beta)$. We actually develop a better stability bound. Specifically, the pointwise hypothesis stability is bounded by $O\big(\frac{1}{n\beta} + \epsilon\big)$ in Charles & Papailiopoulos (2018) while we show that the on-average stability is bounded by $O\big(\frac{1}{n\beta} + \epsilon^2\big)$, which is significantly tighter if $\frac{1}{n\beta} \leq \epsilon \leq 1$ (ignoring constant factors). It should be mentioned that Charles & Papailiopoulos (2018) did not impose a smoothness assumption. However, the smoothness assumption is widely used in non-convex optimization to derive meaningful rates (Ghadimi & Lan, 2013). As compared to probabilistic bounds in Charles & Papailiopoulos (2018), our bounds are stated in expectation. The extension to high-probability bounds will lead to additional $O(1/\sqrt{n})$ term (Feldman & Vondrak, 2019).

**Remark 2** (Bounded gradient assumption). Very recently, the bounded gradient assumption was also removed for the stability analysis (Lei & Ying, 2020). However, their analysis considered SGD applied to convex loss functions. As a comparison, we study stability and generalization in a non-convex learning setting, and our analysis applies to any stochastic optimization algorithms.

**Remark 3.** If $A$ is ERM, Theorem 1 immediately implies $\mathbb{E}\big[F(\mathbf{w}_S) - \hat{F}_S\big] \leq \frac{16L}{n\beta}\mathbb{E}[\hat{F}_S]$. If $F_S$ is $\beta$-strongly convex and $L < n\beta/2$, it was shown for ERM that $\mathbb{E}\big[F(\mathbf{w}_S) - \hat{F}_S\big] \leq \frac{48L}{n\beta}\mathbb{E}\big[\hat{F}_S\big]$ (Shalev-Shwartz & Ben-David, 2014, Corollary 13.7). Their result is extended here from a strongly convex setting to a gradient-dominated setting, and from the particular ERM to any algorithm.

As a direct corollary, we can derive the following optimistic bound in the interpolation setting, which is the most intriguing case for over-parameterized or highly expressive DNN models.

**Corollary 2.** *Let Assumptions 1, 2 hold and* $\mathbf{w}_S = A(S)$. *If* $\mathbb{E}[\hat{F}_S] = 0$ *and* $L < n\beta/2$, *then* $\mathbb{E}\big[F(\mathbf{w}_S)\big] \leq \frac{L}{2\beta}\mathbb{E}\big[F_S(\mathbf{w}_S)\big]$.

**Remark 4.** Corollary 2 shows a benefit of interpolation in boosting the generalization by achieving a generalization bound $O(\epsilon)$ for any $\epsilon > 0$ if we minimize $F_S$ sufficiently well. This benefit can not be explained by the existing discussions (Hardt et al., 2016; Charles & Papailiopoulos, 2018) as they imply the same generalization bound $O(1/\sqrt{n\beta})$ in the interpolation setting. Although it was observed that interpolation helps in training (Bassily et al., 2018; Vaswani et al., 2019; Ma et al., 2018; Oymak & Soltanolkotabi, 2020; Allen-Zhu et al., 2019; Zou et al., 2018), it is still largely unclear, as indicated in Ma et al. (2018), that how interpolation helps in generalization. Corollary 2 shows new insights on how interpolation from highly expressive models helps generalization.

We now move on to the discussion on the critical assumption in Corollary 2, i.e. $L < n\beta/2$. According to the proof, the two parameters $L$ and $\beta$ can be replaced by their local counterparts, i.e., the smoothness and PL condition related to a particular minimizer $\mathbf{w}'$ of $F_{S^{(i)}}$ (Eqs. (B.6), (B.7)). For example, $\beta$ can be replaced by $\frac{1}{2}\|\nabla F_S(\mathbf{w}')\|_2^2/(F_S(\mathbf{w}') - \hat{F}_S)$, which can be larger than $\beta$. Below are some examples on explaining $L/\beta < n/2$. As we will see, the quantity $L/\beta$ reflects the complexity of the problem (related to condition number as shown in Examples 1, 2). Therefore, the condition $L/\beta < n/2$ imposes implicitly a constraint on the complexity of the problems. This explains why the optimization algorithm would never overfit when applied to gradient-dominated objective functions if $L/\beta < n/2$, as shown in Theorem 1.

**Example 1.** Let $\phi : \mathbb{R}^d \mapsto \mathbb{R}^m$ be a feature map, and $\ell : \mathbb{R} \times \mathbb{R} \mapsto \mathbb{R}_+$ be a loss function which is $L_\ell$-smooth and $\sigma_\ell$-strongly convex w.r.t. the first argument. Consider $f(\mathbf{w}; z) = \ell(\langle \mathbf{w}, \phi(x_i)\rangle, y_i)$ with $\langle \cdot, \cdot \rangle$ being an inner product. Then, $F_S$ satisfies the PL condition with the parameter $\sigma'_{\min}(\Sigma_S)\sigma_\ell$, where $\Sigma_S = \frac{1}{n}\sum_{i=1}^n \phi(x_i)\phi(x_i)^\top$ is the empirical covariance matrix, $A^\top$ denotes the transpose of a matrix $A$ and $\sigma'_{\min}(A)$ means the minimal *non-zero* singular value of $A$. The empirical counterpart (we have an expectation w.r.t. $S$ in PL condition) of $L/\beta$ is of the order of $\sigma_{\max}(\Sigma_S)/\sigma'_{\min}(\Sigma_S)$, where $\sigma_{\max}(A)$ means the maximal singular value (we give details in Appendix E.1).

**Example 2.** Consider neural networks with a single hidden layer with $d$ inputs, $m$ hidden neurons and a single output neuron, for which the prediction function takes the form $h_{\mathbf{v},\mathbf{w}} = \sum_{k=1}^{m} v_k \phi(\langle \mathbf{w}_k, x \rangle)$. Here $\mathbf{w}_k \in \mathbb{R}^d$ and $v_k \in \mathbb{R}$ denote the weight of the edges connecting the $k$-th hidden node to the input and output node, respectively, while $\phi : \mathbb{R} \mapsto \mathbb{R}$ is the activation function. Analogous to Arora et al. (2019); Oymak & Soltanolkotabi (2020), we fix $\mathbf{v} = (v_1, \ldots, v_m)^\top$ with $|v_k| = a$ for some $a > 0$ and train $\mathbf{w} = (\mathbf{w}_1, \mathbf{w}_2, \ldots, \mathbf{w}_m)^\top \in \mathbb{R}^{m \times d}$ from $S$. The loss function then takes the form $f(\mathbf{w}; z) = (\mathbf{v}^\top \phi(\mathbf{w}x) - y)^2$. If we consider the identity activation function, i.e., $\phi(t) = t$, then $F_S$ satisfies the PL condition with the parameter $\sigma_{\min}(\Sigma_S)$, where $\sigma_{\min}(A)$ denotes the minimal singular value of $A$ and $\Sigma_S = \frac{1}{n}\sum_{i=1}^{n} x_i x_i^\top$. The empirical counterpart of $L/\beta$ is of the order of $\sigma_{\max}(\Sigma_S)/\sigma_{\min}(\Sigma_S)$ (we give details in Appendix E.2 for a general activation function).

It is possible to get generalization bounds under some other conditions. Since one-point strong convexity condition together with smoothness assumption implies the PL condition (Yuan et al., 2019), all our results apply to one-point strongly convex functions. We can also get generalization bounds for objective functions satisfying the quadratic growth condition (Necoara et al., 2018), which is weaker than the PL condition. However, we need to impose a realizability condition which was also imposed in Charles & Papailiopoulos (2018). The proof of Theorem 3 is given in Section C. Let $\mathbf{w}^{(S)}$ denote the Euclidean projection of $\mathbf{w}$ onto the set of global minimizers of $F_S$ in $\mathcal{W}$.

**Definition 3** (Quadratic Growth Condition). We say $F_S : \mathcal{W} \mapsto \mathbb{R}$ satisfies the quadratic growth condition (in expectation) with parameter $\beta$ if $\mathbb{E}[F_S(\mathbf{w}) - \hat{F}_S] \geq \frac{\beta}{2}\mathbb{E}[\|\mathbf{w} - \mathbf{w}^{(S)}\|_2^2]$ for all $\mathbf{w} \in \mathcal{W}$.

**Theorem 3.** *Let Assumption 1 hold and $F_S$ satisfy the quadratic growth condition with parameter $\beta$. If the problem is realizable, i.e., $\mathbb{E}[\hat{F}_S] = 0$ and $L \leq n\beta/4$, then $\mathbb{E}[F(\mathbf{w}_S)] \leq 2L\beta^{-1}\mathbb{E}[F_S(\mathbf{w}_S)]$.*

Finally, we consider any optimization algorithms applied to gradient-dominated and Lipschitz continuous functions. We do not require loss functions to be smooth here. It shows that the excess generalization bound can decay as fast as $O(1/(n\beta))$ if we solve the optimization problem to a sufficient accuracy, which is much better than the generalization bound $O(1/\sqrt{n\beta})$ in Charles & Papailiopoulos (2018). Recall the analysis in Charles & Papailiopoulos (2018) requires Assumptions 2, 3 and a further assumption on boundedness of loss functions. The proof is given in Section C.

**Theorem 4.** *Let Assumptions 2, 3 hold and $\mathbf{w}_S = A(S)$. Then the following inequality holds*

$$\mathbb{E}[F(\mathbf{w}_S) - \hat{F}_S] \leq \frac{2G^2}{n\beta} + \frac{G(\mathbb{E}[F_S(\mathbf{w}_S) - \hat{F}_S])^{\frac{1}{2}}}{\sqrt{2\beta}}.$$

## 4 APPLICATIONS

In this section, we apply Theorem 1 to different stochastic optimization algorithms such as stochastic gradient descent, randomized coordinate descent, and stochastic variance-reduced optimization. In particular, we study the number of stochastic gradient evaluations required to achieve a prescribed generalization bound, which is summarized in Table 1. We always assume $L \leq n\beta/4$ in this section.

### 4.1 STOCHASTIC GRADIENT DESCENT

We need some notations to state results on SGD. Specifically, denote by $\mathbf{w}_1 \in \mathcal{W}$ an initial point of SGD. At the $t$-th iteration, we first randomly select an index $i_t \sim \text{unif}[n]$, and then update $\{\mathbf{w}_t\}_t$ by

$$\mathbf{w}_{t+1} = \mathbf{w}_t - \eta_t \nabla f(\mathbf{w}_t; z_{i_t}), \tag{4.1}$$

where $\{\eta_t\}_t$ is a sequence of positive step sizes and $\text{unif}[n]$ denotes the uniform distribution over $[n]$. The proof of Theorem 5 is given in Appendix D.1.

**Theorem 5.** *Let Assumptions 1, 2 hold with $L \leq n\beta/4$. Let $A$ be SGD with the step size sequence $\eta_t = \frac{2t+1}{2\beta(t+1)^2}$. Then $\mathbb{E}[F(\mathbf{w}_{T+1})] - F(\mathbf{w}^*) = O\left(\frac{1}{n\beta} + \frac{1}{T\beta^3}\right)$. We can take $O(\frac{n}{\beta^2})$ stochastic gradient evaluations to get excess generalization bounds $O(1/(n\beta))$.*

**Remark 5.** We compare Theorem 5 with the recent generalization analysis of SGD under the PL condition. Based on pointwise hypothesis stability analysis and the optimization error bound in Karimi et al. (2016), it was shown with probability at least $1 - \delta$ (Charles & Papailiopoulos, 2018)

$$F(\mathbf{w}_{T+1}) - F(\mathbf{w}^*) = O\left(\frac{1}{\sqrt{n\beta\delta}} + \frac{1}{T^{\frac{1}{4}}\beta^{\frac{3}{4}}\delta^{\frac{1}{2}}}\right). \tag{4.2}$$

| Algorithm | Complexity for $1/(n\beta)$ | Complexity for $\epsilon$ if $\mathbb{E}[\hat{F}_S] = 0$ |
|---|---|---|
| SGD | $\frac{n}{\beta^2}$ | $\frac{1}{\beta^2} \log \frac{1}{\beta\epsilon}$ |
| RCD | $\frac{d \log n}{\beta}$ | $\frac{d}{\beta} \log \frac{1}{\beta\epsilon}$ |
| SVRG, SCSG | $\left(n + n^{\frac{2}{3}}/\beta\right) \log n$ | $\left(n + n^{\frac{2}{3}}/\beta\right) \log \frac{1}{\beta\epsilon}$ |
| SARAH, SpiderBoost | $\left(n + 1/\beta^2\right) \log n$ | $\left(n + 1/\beta^2\right) \log \frac{1}{\beta\epsilon}$ |
| SNVRG | $\left(n + \sqrt{n}/\beta\right) \log^4 n$ | $\left(n + \sqrt{n}/\beta\right) \log^4 \frac{1}{\beta\epsilon}$ |

Table 1: Iteration complexity for different optimization algorithms to achieve a stated generalization bound under Assumptions 1, 2. In the second column, we present the number of stochastic gradient evaluations to achieve excess generalization bounds $O(1/(n\beta))$. In the third column, we present the number of stochastic gradient evaluations to achieve generalization bounds $O(\epsilon)$ if $\mathbb{E}[\hat{F}_S] = 0$. We ignore constant factors. It is known that variance-reduction techniques improve the iteration complexity to achieve small training errors. Our stability analysis shows that such an improvement is also achieved for testing errors. Note that the stability analysis in Charles & Papailiopoulos (2018) can at most imply an excess generalization bound $O(1/\sqrt{n\beta})$ for these algorithms.

The above bound indicates that $O(n^2/\beta)$ stochastic gradient evaluations are needed to get the excess generalization bounds $O(1/\sqrt{n\beta})$. Based on the uniform stability bound in Hardt et al. (2016) and the optimization error bound in Karimi et al. (2016), it was shown in Yuan et al. (2019) that

$$\mathbb{E}[F(\mathbf{w}_{T+1})] - F(\mathbf{w}^*) = O\left(n^{-1}(\beta T)^{\frac{L/\beta}{1+L/\beta}}\right) + O\left(\frac{1}{T\beta^2}\right). \tag{4.3}$$

By taking an optimal $T = n^{\frac{1+L/\beta}{1+2L/\beta}} \beta^{-\frac{2+3L/\beta}{1+2L/\beta}}$ (ignoring a constant factor) to balance the above two terms, we derive $\mathbb{E}[F(\mathbf{w}_{T+1})] - F(\mathbf{w}^*) = O\left(n^{-\frac{1+L/\beta}{1+2L/\beta}} \beta^{-\frac{L/\beta}{1+2L/\beta}}\right)$. If $L/\beta$ is moderately large, then this bound quickly becomes $\mathbb{E}[F(\mathbf{w}_{T+1})] - F(\mathbf{w}^*) = O(1/\sqrt{n\beta})$. With high probability at least $1 - \delta$, it was shown that SGD with the step size $\eta_t = \frac{c}{(t+2)\log(t+2)}$ gets the bound $F(\mathbf{w}_{T+1}) - F_S(\mathbf{w}_{T+1}) = O\left(\sqrt{c \log T}/\sqrt{n\delta}\right)$ (Zhou et al., 2018b). However, it is not clear how the optimization errors decay with such step sizes. Typically, $c$ should be of the order $O(1/\beta)$ as shown in Karimi et al. (2016) and therefore the stability analysis in Zhou et al. (2018b) can at best achieve the generalization bounds $O\left(\sqrt{\log T}/\sqrt{n\beta}\right)$. To summarize, the existing stability analysis generally implies the generalization bound $O(1/\sqrt{n\beta})$ for SGD in learning with gradient-dominated objectives (Charles & Papailiopoulos, 2018; Zhou et al., 2018b; Yuan et al., 2019), which is significantly improved to $O(1/(n\beta))$ in our paper by the refined stability analysis. It is worth mentioning that, in this comparison, we have used the same optimization error bounds in Karimi et al. (2016), and the analysis in Charles & Papailiopoulos (2018); Zhou et al. (2018b); Yuan et al. (2019) requires a bounded gradient assumption and a bounded loss assumption, which are removed in our analysis.

The above iteration complexity in Theorem 5 can be further improved if we impose a restricted secant inequality (Karimi et al., 2016) on $F_S$, which has been considered for non-convex optimization, e.g., optimizing neural networks (Li & Yuan, 2017). This is a slightly stronger assumption than the PL condition as shown in Karimi et al. (2016).

**Definition 4** (Restricted Secant Inequality). We say $F_S : \mathcal{W} \mapsto \mathbb{R}$ satisfies the restricted secant inequality with parameter $\beta$ if $\mathbb{E}\left[\langle \mathbf{w} - \mathbf{w}^{(S)}, \nabla F_S(\mathbf{w}) \rangle\right] \geq \beta \mathbb{E}[\|\mathbf{w} - \mathbf{w}^{(S)}\|_2^2]$ for all $\mathbf{w} \in \mathcal{W}$.

**Theorem 6.** *Assume $F_S$ satisfies the restricted secant inequality with parameter $\beta$. Let Assumption 1 hold with $L \leq n\beta/4$. Let $A$ be SGD with $\eta_t = 1/(\beta(t+1))$. Then one can take $O(n/\beta)$ stochastic gradient evaluations to achieve the excess generalization bounds $O(1/(n\beta))$.*

Below we apply Theorem 1 to establish fast generalization bounds in an interpolation setting. Our analysis shows that interpolation actually boosts SGD by achieving an exponential convergence of testing errors, which can not be derived from the bound (3.4) in Charles & Papailiopoulos (2018).

**Theorem 7.** *Let Assumptions 1, 2 hold with $L \leq n\beta/4$, and $\mathbb{E}[\hat{F}_S] = 0$. Let $A$ be SGD with $\eta_t = \beta/L^2$. Then $\mathbb{E}[F(\mathbf{w}_{T+1})] \leq \frac{L(1-\beta^2/L^2)^T}{2\beta} \mathbb{E}[F_S(\mathbf{w}_1)]$. We can take $O\left(\beta^{-2} \log(1/(\beta\epsilon))\right)$ stochastic gradient evaluations to achieve the generalization bound $O(\epsilon)$ for any $\epsilon > 0$.*

The above linear convergence does not contradict existing minimax lower bounds where the benefit of interpolation is not considered. The proofs for Theorems 6, 7 are given in Appendix D.1.

**Remark 6.** We discuss some recent work on error bounds in low-noise conditions. Optimization errors of SGD were studied for general non-convex objectives (Vaswani et al., 2019; Ma et al., 2018) and gradient-dominated objectives (Bassily et al., 2018). For binary classification problems with the specific squared loss, it was shown SGD achieves an exponential convergence of testing classification errors under a margin condition, i.e., positive and negative classes are separated by a margin that is strictly positive (Pillaud-Vivien et al., 2018). This was extended to general convex loss functions under the same margin condition (Nitanda & Suzuki, 2019). These discussions consider regularized objective functions (Pillaud-Vivien et al., 2018; Nitanda & Suzuki, 2019), which are strongly convex. The exponential convergence in Pillaud-Vivien et al. (2018); Nitanda & Suzuki (2019) was established for the testing classification errors, i.e., 0-1 loss. As a comparison, we establish an exponential convergence for the testing errors measured by loss functions used in training. In addition, the exponential convergence in Pillaud-Vivien et al. (2018); Nitanda & Suzuki (2019) comes into effect only after a sufficiently large number of iterations, which is not required in Theorem 7.

## 4.2 RANDOMIZED COORDINATE DESCENT

Randomized coordinate descent (RCD) is an efficient optimization algorithm particularly useful for high-dimensional learning problems (Nesterov, 2012). At each iteration it firstly randomly selects a single coordinate $i_t \in \{1, \ldots, d\}$, and then performs the update along the $i_t$-th coordinate as $\mathbf{w}_{t+1} = \mathbf{w}_t - \eta_t \nabla_{i_t} F_S(\mathbf{w}_t) \mathbf{e}_{i_t}$, where $\nabla_i F_S$ denotes the derivative of $F_S$ w.r.t. the $i$-th coordinate and $\mathbf{e}_i$ is a vector in $\mathbb{R}^d$ with the $i$-th coordinate being 1 and other coordinates being 0.

**Theorem 8.** *Let Assumptions 1 and 2 hold with $L \leq n\beta/4$. Let A be RCD with $\eta_t = 1/L$. Then $\mathbb{E}[F(\mathbf{w}_{T+1})] - F(\mathbf{w}^*) = O\left(\frac{1}{n\beta} + \frac{1}{\beta}\left(1 - \frac{\beta}{dL}\right)^T\right)$. We take $O((d \log n)/\beta)$ stochastic gradient evaluations to get excess generalization bounds $O(1/(n\beta))$. If $\mathbb{E}[\hat{F}_S] = 0$, we take $O\left(\beta^{-1} d \log 1/(\beta\epsilon)\right)$ stochastic gradient evaluations to get generalization bounds $O(\epsilon)$ for any $\epsilon > 0$.*

The detailed proof for the above theorem is given in Appendix D.2. As indicated in Remark 1, the discussion in Charles & Papailiopoulos (2018) can only imply the generalization bound $O(1/\sqrt{n\beta})$.

## 4.3 STOCHASTIC VARIANCE-REDUCED OPTIMIZATION

SGD needs a diminishing step size due to the inherent variance of stochastic gradients, which generally yields a sublinear convergence rate (Bottou et al., 2018). Recently, there is a large amount of work to accelerate SGD by using some different gradient estimates with a reduced variance (Johnson & Zhang, 2013; Xiao & Zhang, 2014; Zhang et al., 2013; Allen-Zhu & Hazan, 2016; Fang et al., 2018; Wang et al., 2019; Nguyen et al., 2017; Zhou et al., 2018a; Schmidt et al., 2017; Defazio et al., 2014; Reddi et al., 2016). This class of algorithms proceeds in epochs. Let $\tilde{\mathbf{w}}_0$ be an initialization point. At the beginning of $s$-th epoch, we set a reference point $\mathbf{w}_0 = \tilde{\mathbf{w}}_{s-1}$, draw a batch $\tilde{I}_s \subseteq [n]$ and compute $\mathbf{v}_0 = \nabla f_{\tilde{I}_s}(\mathbf{w}_0)$, where we denote $f_I(\mathbf{w}) = \frac{1}{|I|} \sum_{i \in I} f(\mathbf{w}; z_i)$ for $I \subseteq [n]$ and $|I|$ is the cardinality of $I$. The batch $\tilde{I}_s$ can be equal to $[n]$ (Johnson & Zhang, 2013; Xiao & Zhang, 2014; Wang et al., 2019; Reddi et al., 2016) or drawn with replacement according to the uniform distribution over $[n]$ (Lei et al., 2017; Fang et al., 2018). Then we proceed with $m_s$ inner iterations by using some gradient estimators with reduced variances. At the $t$-th inner iteration, we first draw a batch $I_t \subseteq [n]$ from the uniform distribution over $[n]$. The original SVRG (Johnson & Zhang, 2013; Reddi et al., 2016; Xiao & Zhang, 2014) uses the gradient estimator (we omit the dependency on $s$)
$$\mathbf{v}_t = \nabla f_{I_t}(\mathbf{w}_t) - \nabla f_{I_t}(\mathbf{w}_0) + \mathbf{v}_0. \tag{4.4}$$
Recently a different update of gradient estimator is proposed (Nguyen et al., 2017; Fang et al., 2018)
$$\mathbf{v}_t = \nabla f_{I_t}(\mathbf{w}_t) - \nabla f_{I_t}(\mathbf{w}_{t-1}) + \mathbf{v}_{t-1}. \tag{4.5}$$
An important observation is that the variance of $\mathbf{v}_t$ diminishes to zero as we are approaching the minimum, which allows us to update the iterate with a constant step size $\mathbf{w}_{t+1} = \mathbf{w}_t - \eta \mathbf{v}_t$ (Johnson & Zhang, 2013). The framework of stochastic variance-reduced optimization is described in Algorithm 1 in Appendix D.3. The following theorem gives generalization bounds $O(1/(n\beta))$ for stochastic variance-reduced optimization, which significantly improves the bound $O(1/\sqrt{n\beta})$ based on (3.4). The proof is given in Appendix D.3.

**Theorem 9.** *Let Assumptions 1 and 2 hold with $L \leq n\beta/4$. Let A be either the SARAH in Nguyen et al. (2017) or the SpiderBoost in Wang et al. (2019). We can take $O\left((n + 1/\beta^2) \log n\right)$ stochastic gradient evaluations to get excess generalization bounds $O(1/(n\beta))$. If $\mathbb{E}[\hat{F}_S] = 0$, we take $O\left((n + 1/\beta^2) \log 1/(\beta\epsilon)\right)$ stochastic gradient evaluations to get generalization bounds $O(\epsilon)$ for any $\epsilon > 0$.*

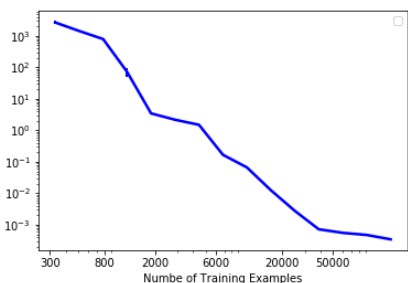

Figure 1: $\kappa_I$ versus $|I|$

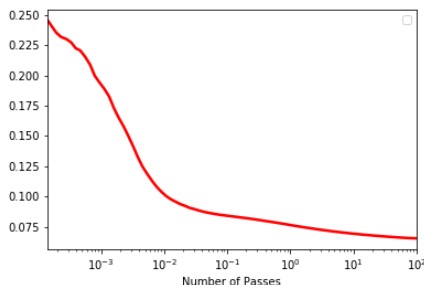

Figure 2: Testing error versus number of passes

As compared to SGD (Section 4.1), Theorem 9 shows SARAH/SpiderBoost requires significantly fewer iterations to achieve the same testing errors. This shows a clear advantage of stochastic variance-reduced optimization over SGD in generalization other than training. Other than SARAH and SpiderBoost, we also develop generalization bounds for SVRG in Reddi et al. (2016), SCSG in Lei et al. (2017) (Theorem D.3) and SNVRG-PL in Zhou et al. (2018a) (Theorem D.4).

# 5 SIMULATIONS AND CONCLUSIONS

**Simulations**. We report some preliminary experiments to support our theory. We consider the dataset IJCNN available from the LIBSVM website (Chang & Lin, 2011) and report the average of experimental results from 25 repetitions. In our first experiment, we aim to check how the condition $\sigma_{\max}(\Sigma_S)/\sigma_{\min}(\Sigma_S) \leq n/4$ would be satisfied in practice. To this aim, we randomly pick a subset $I \subset \{1, 2, \ldots, n\}$ and build an empirical covariance matrix $\Sigma_I = \frac{1}{|I|} \sum_{i \in I} x_i x_i^\top$, where $|I|$ denotes the cardinality of $I$. Then we compute the term $\kappa_I := \frac{\sigma_{\max}(\Sigma_I)}{\sigma_{\min}(\Sigma_I)|I|}$. Figure 1 plots the $\kappa_I$ as a function of $|I|$. It is clear that the condition $\kappa_I \leq 1/4$ is violated if $|I|$ is small. As $|I|$ increases, $\kappa_I$ decreases and can be as small as $10^{-3}$. Then, the condition $\kappa_I \leq 1/4$ holds trivially for sufficiently large $n$.

Theorem 1 implies that overfitting would never happen for learning with gradient-dominated functions. Our second experiment aims to verify this phenomenon. We consider a generalized linear model for binary classification with the loss function $f(\mathbf{w}; z) = \left(\ell(\mathbf{w}^\top x) - y\right)^2$, where $\ell$ is the logistic link function $\ell(a) = (1 + \exp(-a))^{-1}$. It was shown that the corresponding objective function is gradient-dominated (Foster et al., 2018). We use 80 percents of the dataset for training and reserve the remaining 20 percents for testing. We apply SGD with the step size $\eta_t = 1/(1 + 0.001t)$ and compute the testing error of $\{\mathbf{w}_t\}$ on the testing dataset. In Figure 2, we plot the testing errors versus the number of passes (iteration number divided by sample size). It is clear that the testing error continue to decrease along the learning process, and there is no overfitting even after 100 passes of the dataset. This is well consistent with Theorem 1.

**Conclusions**. We study stochastic optimization under the PL condition. We show that the generalization errors can be bounded by $O(1/(n\beta))$ plus the convergence rate of algorithms. An observation is that the optimization always helps in generalization under the PL condition. Our analysis based on a weak on-average stability measure removes the bounded gradient assumption in the literature, and can imply significantly better bounds. In particular, we show how the interpolation accelerates the generalization. Our study relies on an essential PL condition on the objective function. While this assumption is widely used in the non-convex learning setting, it would be very interesting to extend the discussions here to general non-convex objective functions.

## ACKNOWLEDGMENTS

The work of Yunwen Lei is supported by the National Natural Science Foundation of China (Grant No. 61806091) and the Alexander von Humboldt Foundation. The work of Yiming Ying is supported by NSF grants IIS-1816227 and IIS-2008532.

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

## A  STABILITY AND GENERALIZATION

We first give the definition of pointwise hypothesis stability. For any $i \in [n]$, denote $S \backslash z_i = \{z_1, \ldots, z_{i-1}, z_{i+1}, \ldots, z_n\}$.

**Definition 5** (Pointwise Hypothesis Stability). We say a randomized algorithm $A$ has pointwise hypothesis stability $\epsilon$ if for all $i \in [n]$ there holds $\mathbb{E}_{S,A}\big[\big|f(A(S); z_i) - f\big(A(S \backslash z_i); z_i\big)\big|\big] \leq \epsilon$.

Theorem A.1 establishes the key connection between the generalization and various stability measures. Part (a) and part (b) show that the algorithm with either uniform stability or pointwise hypothesis stability generalizes well to testing examples (Bousquet & Elisseeff, 2002). Initially, they were developed for deterministic algorithms (Bousquet & Elisseeff, 2002), which were then extended to the setting of randomized algorithms (Elisseeff et al., 2005). Part (c) shows the connection between the generalization and the on-average stability (Shalev-Shwartz et al., 2010). Note part (b) involves a square root of $1/\delta$ instead of a $\log(1/\delta)$.

**Theorem A.1** (Generalization by Stability). *Let A be a randomized algorithm.*

*(a) If A has uniform stability $\epsilon$, then $\big|\mathbb{E}_{S,A}\big[F_S(A(S)) - F(A(S))\big]\big| \leq \epsilon$.*

*(b) Let $M > 0$. If A has pointwise hypothesis stability $\epsilon$ and $0 \leq f(\mathbf{w}; z) \leq M$ for all $\mathbf{w} \in \mathcal{W}$ and $z \in \mathcal{Z}$. Then for all $\delta \in (0, 1)$ with probability at least $1 - \delta$*

$$F(A(S)) \leq F_S(A(S)) + \Big(\frac{M^2 + 12Mn\epsilon}{n\delta}\Big)^{\frac{1}{2}}. \tag{A.1}$$

*(c) If A has on-average stability $\epsilon$, then $\mathbb{E}_{S,A}\big[F(A(S)) - F_S(A(S))\big] \leq \epsilon$.*

*Proof.* The proof of Part (a) can be found in Hardt et al. (2016, Theorem 2.2). Part (b) was first proved for deterministic algorithms (Bousquet & Elisseeff, 2002, Theorem 11), and then extended to randomized algorithms (Elisseeff et al., 2005, Theorem 12). We prove Part (c) here due to its simplicity. Since $z_i$ and $\tilde{z}_i$ are drawn from the same distribution, we know

$$\mathbb{E}_{S,A}\big[F(A(S)) - F_S(A(S))\big] = \frac{1}{n}\sum_{i=1}^{n} \mathbb{E}_{S,\widetilde{S},A}\big[F(A(S^{(i)})) - F_S(A(S))\big]$$

$$= \frac{1}{n}\sum_{i=1}^{n} \mathbb{E}_{S,\widetilde{S},A}\big[f(A(S^{(i)}); z_i) - f(A(S); z_i)\big],$$

where the last identity holds since $z_i$ is independent of $A(S^{(i)})$. The proof is complete by noting the definition of on-average stability. □

# B  PROOF OF THEOREM 1

In this section, we prove Theorem 1. We begin our analysis with some useful properties of smooth functions. If $g$ is $L$-smooth, we have the following self-bounding property (Srebro et al., 2010)

$$\|\nabla g(\mathbf{w})\|_2^2 \leq 2L\big(g(\mathbf{w}) - \inf_{\mathbf{w}'} g(\mathbf{w}')\big), \quad \forall \mathbf{w} \in \mathcal{W} \tag{B.1}$$

and the following elementary inequality for all $\mathbf{w}, \tilde{\mathbf{w}} \in \mathcal{W}$ (Nesterov, 2012)

$$g(\mathbf{w}) \leq g(\tilde{\mathbf{w}}) + \langle \nabla g(\tilde{\mathbf{w}}), \mathbf{w} - \tilde{\mathbf{w}} \rangle + \frac{L\|\mathbf{w} - \tilde{\mathbf{w}}\|_2^2}{2}. \tag{B.2}$$

In particular, if $g$ is further nonnegative, then

$$\|\nabla g(\mathbf{w})\|_2^2 \leq 2Lg(\mathbf{w}), \quad \forall \mathbf{w} \in \mathcal{W}. \tag{B.3}$$

The following lemma follows directly from the self-bounding property of smooth loss functions.

**Lemma B.1.** *Assume $F$ is $L$-smooth. Then ($\mathbf{w}$ can depend on $S$)*

$$\mathbb{E}[\|\nabla F(\mathbf{w})\|_2^2] \leq 2L\mathbb{E}\big[F(\mathbf{w}) - \hat{F}_S\big].$$

*Proof.* Recall $\mathbf{w}^* = \arg\min_{\mathbf{w} \in \mathcal{W}} F(\mathbf{w})$. According to the self-bounding property (B.1) and the definition of $\mathbf{w}^*$ we know

$$\mathbb{E}[\|\nabla F(\mathbf{w})\|_2^2] \leq 2L\mathbb{E}\big[F(\mathbf{w}) - F(\mathbf{w}^*)\big] = 2L\mathbb{E}\big[F(\mathbf{w}) - F_S(\mathbf{w}^*)\big]$$

$$\leq 2L\mathbb{E}\big[F(\mathbf{w}) - \hat{F}_S\big],$$

where we have used $\mathbb{E}[F_S(\mathbf{w}^*)] = F(\mathbf{w}^*)$ since $\mathbf{w}^*$ is independent of $S$, and $\hat{F}_S \leq F_S(\mathbf{w}^*)$ due to the definition of $\hat{F}_S$. The proof is complete. □

In the following lemma, we derive the on-average stability bounds under the PL condition. Recall for any $\mathbf{w}$, we denote by $\mathbf{w}^{(S)}$ the Euclidean projection of $\mathbf{w}$ onto the set of global minimizers of $F_S$ in $\mathcal{W}$.

**Lemma B.2.** *If Assumptions 1, 2 hold, then A has on-average stability $\epsilon$ satisfying*

$$\epsilon \leq \frac{2L}{n\beta}\Big(\mathbb{E}[\hat{F}_S] + \mathbb{E}[F(\mathbf{w}_S^{(S)})]\Big) + \mathbb{E}\big[F(\mathbf{w}_S) - F(\mathbf{w}_S^{(S)})\big] + \mathbb{E}\big[\hat{F}_S - F_S(\mathbf{w}_S)\big].$$

*Proof.* Let $\widetilde{S} = \{\tilde{z}_1, \ldots, \tilde{z}_n\}$ be drawn independently from $\rho$. For each $i \in [n]$, let $S^{(i)}$ be defined in Definition 2. For each $i \in [n]$, we denote $\mathbf{w}_{S^{(i)}} = A(S^{(i)})$ and $\mathbf{w}_{S^{(i)}}^{(S^{(i)})}$ the projection of $\mathbf{w}_{S^{(i)}}$ onto the set of global minimizer of $F_{S^{(i)}}$. We decompose $f(\mathbf{w}_{S^{(i)}}; z_i) - f(\mathbf{w}_S; z_i)$ as follows

$$
\begin{aligned}
f(\mathbf{w}_{S^{(i)}}; z_i) - f(\mathbf{w}_S; z_i) = \big( f(\mathbf{w}_{S^{(i)}}; z_i) - f(\mathbf{w}_{S^{(i)}}^{(S^{(i)})}; z_i) \big) \\
+ \big( f(\mathbf{w}_{S^{(i)}}^{(S^{(i)})}; z_i) - f(\mathbf{w}_S^{(S)}; z_i) \big) + \big( f(\mathbf{w}_S^{(S)}; z_i) - f(\mathbf{w}_S; z_i) \big).
\end{aligned} \quad \text{(B.4)}
$$

We now address the above three terms separately. We **first** address $f(\mathbf{w}_{S^{(i)}}^{(S^{(i)})}; z_i) - f(\mathbf{w}_S^{(S)}; z_i)$. According to the definition of $F_S, S, S^{(i)}$, we know

$$
f(\mathbf{w}_{S^{(i)}}^{(S^{(i)})}; z_i) = n F_S(\mathbf{w}_{S^{(i)}}^{(S^{(i)})}) - n F_{S^{(i)}}(\mathbf{w}_{S^{(i)}}^{(S^{(i)})}) + f(\mathbf{w}_{S^{(i)}}^{(S^{(i)})}; \tilde{z}_i).
$$

Since $z_i$ and $\tilde{z}_i$ follow from the same distribution, we know $\mathbb{E}[f(\mathbf{w}_{S^{(i)}}^{(S^{(i)})}; \tilde{z}_i)] = \mathbb{E}[f(\mathbf{w}_S^{(S)}; z_i)]$ and further get

$$
\mathbb{E}\big[f(\mathbf{w}_{S^{(i)}}^{(S^{(i)})}; z_i)\big] = n\mathbb{E}\big[F_S(\mathbf{w}_{S^{(i)}}^{(S^{(i)})})\big] - n\mathbb{E}\big[F_{S^{(i)}}(\mathbf{w}_{S^{(i)}}^{(S^{(i)})})\big] + \mathbb{E}\big[f(\mathbf{w}_S^{(S)}; z_i)\big].
$$

It then follows that

$$
\begin{aligned}
\mathbb{E}\big[f(\mathbf{w}_{S^{(i)}}^{(S^{(i)})}; z_i) - f(\mathbf{w}_S^{(S)}; z_i)\big] &= n\mathbb{E}\big[F_S(\mathbf{w}_{S^{(i)}}^{(S^{(i)})}) - F_{S^{(i)}}(\mathbf{w}_{S^{(i)}}^{(S^{(i)})})\big] \\
&= n\mathbb{E}\Big[F_S(\mathbf{w}_{S^{(i)}}^{(S^{(i)})}) - \inf_{\mathbf{w}\in\mathcal{W}} F_S(\mathbf{w})\Big], \quad \text{(B.5)}
\end{aligned}
$$

where we have used the following identity due to the symmetry between $z_i$ and $\tilde{z}_i$

$$
\mathbb{E}[F_{S^{(i)}}(\mathbf{w}_{S^{(i)}}^{(S^{(i)})})] = \mathbb{E}[\hat{F}_S] = \mathbb{E}\big[\inf_{\mathbf{w}\in\mathcal{W}} F_S(\mathbf{w})\big].
$$

By the PL condition of $F_S$, it then follows from (B.5) that (in our assumption of PL condition, $\mathbf{w}$ may depend on $S$. This was also imposed in the literature (Yuan et al., 2019; Charles & Papailiopoulos, 2018; Zhou et al., 2018b). Indeed, the PL condition was often shown for empirical functions $F_S$)

$$
\mathbb{E}\big[f(\mathbf{w}_{S^{(i)}}^{(S^{(i)})}; z_i) - f(\mathbf{w}_S^{(S)}; z_i)\big] \le \frac{n}{2\beta}\mathbb{E}\big[\|\nabla F_S(\mathbf{w}_{S^{(i)}}^{(S^{(i)})})\|_2^2\big]. \quad \text{(B.6)}
$$

According to the definition of $\mathbf{w}_{S^{(i)}}^{(S^{(i)})}$ we know $\nabla F_{S^{(i)}}(\mathbf{w}_{S^{(i)}}^{(S^{(i)})}) = 0$ and therefore $((a+b)^2 \le 2a^2 + 2b^2)$

$$
\begin{aligned}
\|\nabla F_S(\mathbf{w}_{S^{(i)}}^{(S^{(i)})})\|_2^2 &= \Big\|\nabla F_{S^{(i)}}(\mathbf{w}_{S^{(i)}}^{(S^{(i)})}) - \frac{1}{n}\nabla f(\mathbf{w}_{S^{(i)}}^{(S^{(i)})}; \tilde{z}_i) + \frac{1}{n}\nabla f(\mathbf{w}_{S^{(i)}}^{(S^{(i)})}; z_i)\Big\|_2^2 \\
&\le \frac{2}{n^2}\|\nabla f(\mathbf{w}_{S^{(i)}}^{(S^{(i)})}; \tilde{z}_i)\|_2^2 + \frac{2}{n^2}\|\nabla f(\mathbf{w}_{S^{(i)}}^{(S^{(i)})}; z_i)\|_2^2 \\
&\le \frac{4L}{n^2} f(\mathbf{w}_{S^{(i)}}^{(S^{(i)})}; \tilde{z}_i) + \frac{4L}{n^2} f(\mathbf{w}_{S^{(i)}}^{(S^{(i)})}; z_i), \quad \text{(B.7)}
\end{aligned}
$$

where we have used the self-bounding property of smooth loss functions (B.3). Since $z_i$ and $\tilde{z}_i$ follow from the same distribution, we know

$$
\mathbb{E}[f(\mathbf{w}_{S^{(i)}}^{(S^{(i)})}; \tilde{z}_i)] = \mathbb{E}[f(\mathbf{w}_S^{(S)}; z_i)], \qquad \mathbb{E}[f(\mathbf{w}_{S^{(i)}}^{(S^{(i)})}; z_i)] = \mathbb{E}[f(\mathbf{w}_S^{(S)}; \tilde{z}_i)].
$$

It then follows that

$$
\mathbb{E}\big[\|\nabla F_S(\mathbf{w}_{S^{(i)}}^{(S^{(i)})})\|_2^2\big] \le \frac{4L}{n^2}\mathbb{E}[f(\mathbf{w}_S^{(S)}; z_i)] + \frac{4L}{n^2}\mathbb{E}[f(\mathbf{w}_S^{(S)}; \tilde{z}_i)],
$$

which, combined with (B.6), gives

$$
\mathbb{E}\big[f(\mathbf{w}_{S^{(i)}}^{(S^{(i)})}; z_i) - f(\mathbf{w}_S^{(S)}; z_i)\big] \le \frac{2L}{n\beta}\Big(\mathbb{E}[f(\mathbf{w}_S^{(S)}; z_i)] + \mathbb{E}[f(\mathbf{w}_S^{(S)}; \tilde{z}_i)]\Big).
$$

Taking a summation of the above inequality for $i = 1, \ldots, n$, we get

$$
\sum_{i=1}^n \mathbb{E}\big[f(\mathbf{w}_{S^{(i)}}^{(S^{(i)})}; z_i) - f(\mathbf{w}_S^{(S)}; z_i)\big] \le \frac{2L}{\beta}\Big(\mathbb{E}[\hat{F}_S] + \mathbb{E}[F_{\widetilde{S}}(\mathbf{w}_S^{(S)})]\Big). \quad \text{(B.8)}
$$

We **then** address $f(\mathbf{w}_{S^{(i)}}; z_i) - f(\mathbf{w}_{S^{(i)}}^{(S^{(i)})}; z_i)$. Since $\mathbf{w}_{S^{(i)}}$ and $\mathbf{w}_{S^{(i)}}^{(S^{(i)})}$ are independent of $z_i$, we know

$$
\mathbb{E}\big[f(\mathbf{w}_{S^{(i)}}; z_i) - f(\mathbf{w}_{S^{(i)}}^{(S^{(i)})}; z_i)\big] = \mathbb{E}\big[F(\mathbf{w}_{S^{(i)}}) - F(\mathbf{w}_{S^{(i)}}^{(S^{(i)})})\big] = \mathbb{E}\big[F(\mathbf{w}_S) - F(\mathbf{w}_S^{(S)})\big], \quad \text{(B.9)}
$$

where we have used the symmetry between $z_i$ and $\tilde{z}_i$.

**Finally**, we address $f(\mathbf{w}_S^{(S)}; z_i) - f(\mathbf{w}_S; z_i)$. By the definition of $\mathbf{w}_S^{(S)}$ we know

$$\sum_{i=1}^{n} \left( f(\mathbf{w}_S^{(S)}; z_i) - f(\mathbf{w}_S; z_i) \right) = n\left( \hat{F}_S - F_S(\mathbf{w}_S) \right). \tag{B.10}$$

Plugging (B.8), (B.9) and the above inequality back into (B.4), we derive

$$\sum_{i=1}^{n} \mathbb{E}\left[ f(\mathbf{w}_{S^{(i)}}; z_i) - f(\mathbf{w}_S; z_i) \right] \leq \frac{2L}{\beta} \left( \mathbb{E}[\hat{F}_S] + \mathbb{E}[F_{\widetilde{S}}(\mathbf{w}_S^{(S)})] \right) +$$
$$n\mathbb{E}\left[ F(\mathbf{w}_S) - F(\mathbf{w}_S^{(S)}) \right] + n\mathbb{E}\left[ \hat{F}_S - F_S(\mathbf{w}_S) \right].$$

The proof is complete by recalling the definition of on-average stability and $\mathbb{E}[F_{\widetilde{S}}(\mathbf{w}_S^{(S)})] = \mathbb{E}[F(\mathbf{w}_S^{(S)})]$. $\qquad\square$

We further require a lemma relating the convergence in terms of function values to the convergence in terms of models. This shows that the PL condition is stronger than a quadratic growth condition (Karimi et al., 2016).

**Lemma B.3** (Karimi et al. 2016). *If $F_S$ satisfies the PL condition with parameter $\beta > 0$. Then for all $\mathbf{w} \in \mathcal{W}$ we have*

$$\mathbb{E}\left[ F_S(\mathbf{w}) - F_S(\mathbf{w}^{(S)}) \right] \geq 2\beta \mathbb{E}[\|\mathbf{w} - \mathbf{w}^{(S)}\|_2^2]. \tag{B.11}$$

We are now in a position to prove Theorem 1.

*Proof of Theorem 1.* Plugging the on-average stability established in Lemma B.2 back into Part (c) of Theorem A.1, we derive

$$\mathbb{E}\left[ F(\mathbf{w}_S) - F_S(\mathbf{w}_S) \right] \leq \frac{2L}{n\beta} \left( \mathbb{E}[\hat{F}_S] + \mathbb{E}[F(\mathbf{w}_S^{(S)})] \right) +$$
$$\mathbb{E}\left[ F(\mathbf{w}_S) - F(\mathbf{w}_S^{(S)}) \right] + \mathbb{E}\left[ \hat{F}_S - F_S(\mathbf{w}_S) \right], \tag{B.12}$$

from which we derive

$$\mathbb{E}\left[ F(\mathbf{w}_S^{(S)}) - \hat{F}_S \right] \leq \frac{2L}{n\beta} \left( \mathbb{E}[\hat{F}_S] + \mathbb{E}[F(\mathbf{w}_S^{(S)})] \right). \tag{B.13}$$

By (B.2), we know the following inequality for all $\gamma > 0$

$$F(\mathbf{w}_S) - F(\mathbf{w}_S^{(S)}) \leq \langle \nabla F(\mathbf{w}_S^{(S)}), \mathbf{w}_S - \mathbf{w}_S^{(S)} \rangle + \frac{L}{2}\|\mathbf{w}_S - \mathbf{w}_S^{(S)}\|_2^2$$
$$\leq \|\nabla F(\mathbf{w}_S^{(S)})\|_2 \|\mathbf{w}_S - \mathbf{w}_S^{(S)}\|_2 + \frac{L}{2}\|\mathbf{w}_S - \mathbf{w}_S^{(S)}\|_2^2$$
$$\leq \frac{1}{4\gamma}\|\nabla F(\mathbf{w}_S^{(S)})\|_2^2 + \left( \gamma + \frac{L}{2} \right)\|\mathbf{w}_S - \mathbf{w}_S^{(S)}\|_2^2,$$

where we have used the Cauchy-Schwartz inequality. This together with Lemma B.1 with $\mathbf{w} = \mathbf{w}_S^{(S)}$ implies that

$$\mathbb{E}[F(\mathbf{w}_S) - F(\mathbf{w}_S^{(S)})] \leq \frac{L}{2\gamma}\mathbb{E}\left[ F(\mathbf{w}_S^{(S)}) - \hat{F}_S \right] + \left( \gamma + \frac{L}{2} \right)\mathbb{E}[\|\mathbf{w}_S - \mathbf{w}_S^{(S)}\|_2^2].$$

Plugging (B.13) into the above inequality, we get

$$\mathbb{E}\left[ F(\mathbf{w}_S) - F(\mathbf{w}_S^{(S)}) \right] \leq \frac{L}{2\gamma}\frac{2L}{n\beta} \left( \mathbb{E}[\hat{F}_S] + \mathbb{E}[F(\mathbf{w}_S^{(S)})] \right) + \left( \gamma + \frac{L}{2} \right)\mathbb{E}[\|\mathbf{w}_S - \mathbf{w}_S^{(S)}\|_2^2].$$

Taking $\gamma = L/2$, we then get

$$\mathbb{E}\left[ F(\mathbf{w}_S) - F(\mathbf{w}_S^{(S)}) \right] \leq \frac{2L}{n\beta} \left( \mathbb{E}[\hat{F}_S] + \mathbb{E}[F(\mathbf{w}_S^{(S)})] \right) + L\mathbb{E}[\|\mathbf{w}_S - \mathbf{w}_S^{(S)}\|_2^2].$$

Plugging the above inequality back into (B.12), we derive the following inequality

$$\mathbb{E}\left[ F(\mathbf{w}_S) - F_S(\mathbf{w}_S) \right] \leq \frac{4L}{n\beta} \left( \mathbb{E}[\hat{F}_S] + \mathbb{E}[F(\mathbf{w}_S^{(S)})] \right) + L\mathbb{E}[\|\mathbf{w}_S - \mathbf{w}_S^{(S)}\|_2^2] + \mathbb{E}\left[ \hat{F}_S - F_S(\mathbf{w}_S) \right].$$

It then follows that

$$\mathbb{E}\left[ F(\mathbf{w}_S) - \hat{F}_S \right] \leq \frac{4L}{n\beta} \left( \mathbb{E}[\hat{F}_S] + \mathbb{E}[F(\mathbf{w}_S^{(S)})] \right) + L\mathbb{E}[\|\mathbf{w}_S - \mathbf{w}_S^{(S)}\|_2^2]. \tag{B.14}$$

Since $L \leq n\beta/4$, it follows from (B.13) that

$$\mathbb{E}\big[F(\mathbf{w}_S^{(S)}) - \hat{F}_S\big] \leq \frac{1}{2}\Big(\mathbb{E}[\hat{F}_S] + \mathbb{E}[F(\mathbf{w}_S^{(S)})]\Big)$$

and therefore

$$\mathbb{E}\big[F(\mathbf{w}_S^{(S)})\big] \leq 3\mathbb{E}[\hat{F}_S].$$

We can plug the above inequality back into (B.14) and derive

$$\mathbb{E}\big[F(\mathbf{w}_S) - \hat{F}_S\big] \leq \frac{16L\mathbb{E}[\hat{F}_S]}{n\beta} + L\mathbb{E}[\|\mathbf{w}_S - \mathbf{w}_S^{(S)}\|_2^2]. \tag{B.15}$$

The stated bound then follows from (B.11). The proof is complete. $\qquad\square$

Our analysis in the proof of Theorem 1 actually gives

$$\mathbb{E}\big[F(\mathbf{w}_S) - \hat{F}_S\big] \leq \frac{8L\mathbb{E}[\hat{F}_S]}{n\beta - 2L} + \frac{L\mathbb{E}\big[F_S(\mathbf{w}_S) - \hat{F}_S\big]}{2\beta}.$$

Since we assume $\mathbb{E}[\hat{F}_S] = 0$ in Corollary 2, we only need the condition $L < n\beta/2$ to get Corollary 2.

## C    PROOF OF THEOREM 3 AND THEOREM 4

In this section, we present the proof of Theorem 3 and Theorem 4.

*Proof of Theorem 3.* Let $\tilde{\mathbf{w}}$ be the projection of $\mathbf{w}_{S^{(i)}}^{(S^{(i)})}$ onto the set of global minimizer of $F_S$. Then by the quadratic growth condition, we know

$$\mathbb{E}\big[F_S(\mathbf{w}_{S^{(i)}}^{(S^{(i)})}) - \hat{F}_S\big] \geq \frac{\beta}{2}\mathbb{E}\big[\|\mathbf{w}_{S^{(i)}}^{(S^{(i)})} - \tilde{\mathbf{w}}\|_2^2\big].$$

This together with (B.5) and non-negativity of $f$ implies

$$\frac{n\beta}{2}\mathbb{E}\big[\|\mathbf{w}_{S^{(i)}}^{(S^{(i)})} - \tilde{\mathbf{w}}\|_2^2\big] \leq \mathbb{E}\big[f(\mathbf{w}_{S^{(i)}}^{(S^{(i)})}; z_i)\big] = \mathbb{E}\big[F(\mathbf{w}_{S^{(i)}}^{(S^{(i)})})\big] = \mathbb{E}\big[F(\mathbf{w}_S^{(S)})\big], \tag{C.1}$$

where we have used the symmetry between $S$ and $S^{(i)}$. By the realizability condition, we know almost surely that

$$f(\mathbf{w}_S^{(S)}; z_i) = f(\tilde{\mathbf{w}}; z_i) = 0$$

and $\nabla f(\tilde{\mathbf{w}}; z_i) = 0$. It then follows from the smoothness assumption that

$$\mathbb{E}\big[f(\mathbf{w}_{S^{(i)}}^{(S^{(i)})}; z_i) - f(\mathbf{w}_S^{(S)}; z_i)\big] = \mathbb{E}\big[f(\mathbf{w}_{S^{(i)}}^{(S^{(i)})}; z_i) - f(\tilde{\mathbf{w}}; z_i)\big]$$

$$\leq \mathbb{E}\big[\langle \mathbf{w}_{S^{(i)}}^{(S^{(i)})} - \tilde{\mathbf{w}}, \nabla f(\tilde{\mathbf{w}}; z_i)\rangle + \frac{L}{2}\|\mathbf{w}_{S^{(i)}}^{(S^{(i)})} - \tilde{\mathbf{w}}\|_2^2\big]$$

$$= \mathbb{E}\big[\frac{L}{2}\|\mathbf{w}_{S^{(i)}}^{(S^{(i)})} - \tilde{\mathbf{w}}\|_2^2\big] \leq \frac{L\mathbb{E}[F(\mathbf{w}_S^{(S)})]}{n\beta}.$$

We can plug (B.9), (B.10) and the above inequality back into (B.4), and derive the following bound on the on-average stability $\epsilon$

$$\epsilon \leq \frac{L\mathbb{E}[F(\mathbf{w}_S^{(S)})]}{n\beta} + \mathbb{E}\big[F(\mathbf{w}_S) - F(\mathbf{w}_S^{(S)})\big] + \mathbb{E}\big[\hat{F}_S - F_S(\mathbf{w}_S)\big].$$

We then can analyze analogously to the proof of Theorem 1 but using the above stability bound and get the stated generalization bound. The proof is complete. $\qquad\square$

*Proof of Theorem 4.* Similar to (B.7) but using the boundedness of gradients, we know

$$\|\nabla F_S(\mathbf{w}_{S^{(i)}}^{(S^{(i)})})\|_2^2 \leq \frac{4G^2}{n^2}.$$

We can plug this inequality into (B.6) and derive

$$\mathbb{E}\big[f(\mathbf{w}_{S^{(i)}}^{(S^{(i)})}; z_i) - f(\mathbf{w}_S^{(S)}; z_i)\big] \leq \frac{n}{2\beta}\frac{4G^2}{n^2} = \frac{2G^2}{n\beta}.$$

Taking a summation of the above inequality gives

$$\sum_{i=1}^{n} \mathbb{E}\big[f(\mathbf{w}_{S^{(i)}}^{(S^{(i)})}; z_i) - f(\mathbf{w}_S^{(S)}; z_i)\big] \leq 2G^2/\beta. \tag{C.2}$$

Plugging (C.2), (B.9) and (B.10) back into (B.4), we derive the following inequality

$$\sum_{i=1}^{n} \mathbb{E}\big[f(\mathbf{w}_{S^{(i)}}; z_i) - f(\mathbf{w}_S; z_i)\big] \leq \frac{2G^2}{\beta} + n\mathbb{E}\big[F(\mathbf{w}_S) - F(\mathbf{w}_S^{(S)})\big] + n\mathbb{E}\big[\hat{F}_S - F_S(\mathbf{w}_S)\big]$$

$$\leq \frac{2G^2}{\beta} + nG\mathbb{E}\big[\|\mathbf{w}_S - \mathbf{w}_S^{(S)}\|_2\big] + n\mathbb{E}\big[\hat{F}_S - F_S(\mathbf{w}_S)\big],$$

where in the last step we have used the inequality $F(\mathbf{w}_S) - F(\mathbf{w}_S^{(S)}) \leq G\|\mathbf{w}_S - \mathbf{w}_S^{(S)}\|_2$ due to the boundedness of gradients. According to the definition of on-average stability, we know that the on-average stability $\epsilon$ of $A$ satisfies

$$\epsilon \leq \frac{2G^2}{n\beta} + G\mathbb{E}\big[\|\mathbf{w}_S - \mathbf{w}_S^{(S)}\|_2\big] + \mathbb{E}\big[\hat{F}_S - F_S(\mathbf{w}_S)\big]$$

$$\leq \frac{2G^2}{n\beta} + \frac{G\big(\mathbb{E}\big[F_S(\mathbf{w}_S) - \hat{F}_S\big]\big)^{\frac{1}{2}}}{\sqrt{2\beta}} + \mathbb{E}\big[\hat{F}_S - F_S(\mathbf{w}_S)\big],$$

where we have used Lemma B.3. According to Part (c) of Theorem A.1, it follows that

$$\mathbb{E}\big[F(\mathbf{w}_S) - F_S(\mathbf{w}_S)\big] \leq \frac{2G^2}{n\beta} + \frac{G\big(\mathbb{E}\big[F_S(\mathbf{w}_S) - \hat{F}_S\big]\big)^{\frac{1}{2}}}{\sqrt{2\beta}} + \mathbb{E}\big[\hat{F}_S - F_S(\mathbf{w}_S)\big].$$

The stated bound then follows directly. The proof is complete. □

# D    PROOFS ON APPLICATIONS

In this section, we prove generalization bounds for various stochastic optimization algorithms.

## D.1    STOCHASTIC GRADIENT DESCENT

We consider here SGD. In the following proposition, we establish the variance of stochastic gradients for SGD under the PL condition. The variance was also studied in a general nonconvex setting (Lei et al., 2020a).

**Proposition D.1.** *Let Assumptions 1, 2 hold. Let $\{\mathbf{w}_t\}_t$ be the sequence produced by SGD with step size sequence $\{\eta_t\}_{t\in\mathbb{N}}$. If there exists $t_0 \in \mathbb{N}$ such that $\eta_t \leq \beta/L^2$ for all $t \geq t_0$, then*
$$\mathbb{E}[\|\nabla f(\mathbf{w}_t; z_{i_t})\|_2^2] \leq 2L\max\{\mathbb{E}[F_S(\mathbf{w}_{t_0})], 2\mathbb{E}[\hat{F}_S]\} \quad \forall t \geq t_0.$$

*Proof.* By (B.2) and the update (4.1), we know

$$F_S(\mathbf{w}_{t+1}) \leq F_S(\mathbf{w}_t) + \langle \mathbf{w}_{t+1} - \mathbf{w}_t, \nabla F_S(\mathbf{w}_t)\rangle + \frac{L\|\mathbf{w}_{t+1} - \mathbf{w}_t\|_2^2}{2}$$

$$= F_S(\mathbf{w}_t) - \eta_t\langle \nabla f(\mathbf{w}_t; z_{i_t}), \nabla F_S(\mathbf{w}_t)\rangle + \frac{L\eta_t^2\|\nabla f(\mathbf{w}_t; z_{i_t})\|_2^2}{2}$$

$$\leq F_S(\mathbf{w}_t) - \eta_t\langle \nabla f(\mathbf{w}_t; z_{i_t}), \nabla F_S(\mathbf{w}_t)\rangle + L^2\eta_t^2 f(\mathbf{w}_t; z_{i_t}),$$

where we have used (B.3). Taking expectations on both sides we get the following inequality for all $t \geq t_0$

$$\mathbb{E}[F_S(\mathbf{w}_{t+1})] \leq \mathbb{E}[F_S(\mathbf{w}_t)] - \eta_t\mathbb{E}[\|\nabla F_S(\mathbf{w}_t)\|_2^2] + L^2\eta_t^2\mathbb{E}[f(\mathbf{w}_t; z_{i_t})]$$

$$\leq \mathbb{E}[F_S(\mathbf{w}_t)] - 2\eta_t\beta\mathbb{E}[F_S(\mathbf{w}_t) - \hat{F}_S] + \eta_t\beta\mathbb{E}[F_S(\mathbf{w}_t)], \tag{D.1}$$

where we have used the PL condition and $\eta_t \leq \beta/L^2$ in the last step. It then follows the following inequality for all $t \geq t_0$

$$\mathbb{E}[F_S(\mathbf{w}_{t+1})] \leq (1 - \eta_t\beta)\mathbb{E}[F_S(\mathbf{w}_t)] + \eta_t\beta \cdot 2\mathbb{E}[\hat{F}_S] \leq \max\big\{\mathbb{E}[F_S(\mathbf{w}_t)], 2\mathbb{E}[\hat{F}_S]\big\}.$$

Applying this inequality recursively, we derive

$$\mathbb{E}[F_S(\mathbf{w}_{t+1})] \leq \max\{\mathbb{E}[F_S(\mathbf{w}_{t_0})], 2\mathbb{E}[\hat{F}_S]\} \quad \forall t \geq t_0.$$

This together with (B.3) implies the following inequality for all $t \geq t_0$

$$\mathbb{E}[\|\nabla f(\mathbf{w}_t; z_{i_t})\|_2^2] \leq 2L\mathbb{E}[f(\mathbf{w}_t; z_{i_t})] \leq 2L\max\{\mathbb{E}[F_S(\mathbf{w}_{t_0})], 2\mathbb{E}[\hat{F}_S]\}.$$

The proof is complete. □

We now prove generalization bounds in Theorem 5. We denote $B \asymp \widetilde{B}$ if there exist some constants $c_1$ and $c_2 > 0$ such that $c_1 B \leq \widetilde{B} \leq c_2 B$.

*Proof of Theorem 5.* Let $t_0 = \lfloor L^2/\beta^2 \rfloor$. It is clear that $\eta_t \leq \beta/L^2$ for all $t \geq t_0$. Let $\sigma = 2L \max\{\mathbb{E}[F_S(\mathbf{w}_1)], \ldots, \mathbb{E}[F_S(\mathbf{w}_{t_0})], 2\mathbb{E}[\hat{F}_S]\}$. According to the self-bounding property (B.3) and Proposition D.1, we know that $\mathbb{E}[\|\nabla f(\mathbf{w}_t; z_{i_t})\|_2^2] \leq \sigma^2$ for all $t \in \mathbb{N}$. The following optimization error bound was established in Karimi et al. (2016)

$$\mathbb{E}[F_S(\mathbf{w}_{t+1}) - \hat{F}_S] \leq \frac{L\sigma^2}{2t\beta^2}. \tag{D.2}$$

We can plug the above inequality into (3.2) with $A(S) = \mathbf{w}_{T+1}$, and get

$$\mathbb{E}[F(\mathbf{w}_{T+1}) - \hat{F}_S] \leq \frac{16L\mathbb{E}[\hat{F}_S]}{n\beta} + \frac{L^2\sigma^2}{4T\beta^3}.$$

Since

$$\mathbb{E}[\hat{F}_S] \leq \mathbb{E}[F_S(\mathbf{w}^*)] = F(\mathbf{w}^*), \tag{D.3}$$

we further get

$$\mathbb{E}[F(\mathbf{w}_{T+1})] - F(\mathbf{w}^*) \leq \frac{16LF(\mathbf{w}^*)}{n\beta} + \frac{L^2\sigma^2}{4T\beta^3}.$$

By taking $T \asymp n/\beta^2$, we get $\mathbb{E}[F(\mathbf{w}_{T+1}) - \hat{F}_S] = O(1/(n\beta))$. This corresponds to $O(n/\beta^2)$ stochastic gradient evaluations. The proof is complete. $\qquad \square$

**Lemma D.2.** *Assume $F_S$ satisfies the restricted secant inequality with parameter $\beta$. Let $A$ be SGD with the step size sequence $\eta_t = 1/(\beta(t+1))$. Then there exists some $\sigma \in \mathbb{R}$ such that*
$$\mathbb{E}[\|\mathbf{w}_T - \mathbf{w}_T^{(S)}\|_2^2] \leq \sigma^2/(\beta^2 T).$$

*Proof of Lemma D.2.* Analogous to the proof of Theorem 5, we can find $\sigma \in \mathbb{R}_+$ such that $\mathbb{E}[\|\nabla f(\mathbf{w}_t; z_{i_t})\|_2^2] \leq \sigma^2$ for all $t \in \mathbb{N}$. Since $\mathbf{w}_{t+1}^{(S)}$ is a projection of $\mathbf{w}_{t+1}$ onto the set of global minimizer of $F_S$, we know

$$\|\mathbf{w}_{t+1} - \mathbf{w}_{t+1}^{(S)}\|_2^2 \leq \|\mathbf{w}_{t+1} - \mathbf{w}_t^{(S)}\|_2^2 = \|\mathbf{w}_t - \eta_t \nabla f(\mathbf{w}_t; z_{i_t}) - \mathbf{w}_t^{(S)}\|_2^2$$

$$= \|\mathbf{w}_t - \mathbf{w}_t^{(S)}\|_2^2 + \eta_t^2 \|\nabla f(\mathbf{w}_t; z_{i_t})\|_2^2 + 2\eta_t \langle \mathbf{w}_t^{(S)} - \mathbf{w}_t, \nabla f(\mathbf{w}_t; z_{i_t}) \rangle.$$

Taking an expectation and using $\mathbb{E}[\|\nabla f(\mathbf{w}_t; z_{i_t})\|_2^2] \leq \sigma^2$, we derive

$$\mathbb{E}[\|\mathbf{w}_{t+1} - \mathbf{w}_{t+1}^{(S)}\|_2^2] \leq \mathbb{E}[\|\mathbf{w}_t - \mathbf{w}_t^{(S)}\|_2^2] + \eta_t^2 \sigma^2 + 2\eta_t \mathbb{E}[\langle \mathbf{w}_t^{(S)} - \mathbf{w}_t, \nabla F_S(\mathbf{w}_t) \rangle]$$

$$\leq \mathbb{E}[\|\mathbf{w}_t - \mathbf{w}_t^{(S)}\|_2^2] + \eta_t^2 \sigma^2 - 2\eta_t \beta \mathbb{E}[\|\mathbf{w}_t - \mathbf{w}_t^{(S)}\|_2^2]$$

$$= (1 - 2\eta_t \beta)\mathbb{E}[\|\mathbf{w}_t - \mathbf{w}_t^{(S)}\|_2^2] + \eta_t^2 \sigma^2,$$

where we have used the restricted secant inequality. For the step size $\eta_t = 1/(\beta(t+1))$, we have

$$\mathbb{E}[\|\mathbf{w}_{t+1} - \mathbf{w}_{t+1}^{(S)}\|_2^2] \leq \frac{t-1}{t+1}\mathbb{E}[\|\mathbf{w}_t - \mathbf{w}_t^{(S)}\|_2^2] + \frac{\sigma^2}{\beta^2(t+1)^2}.$$

Multiplying both sides by $t(t+1)$, we derive

$$t(t+1)\mathbb{E}[\|\mathbf{w}_{t+1} - \mathbf{w}_{t+1}^{(S)}\|_2^2] \leq (t-1)t\mathbb{E}[\|\mathbf{w}_t - \mathbf{w}_t^{(S)}\|_2^2] + \frac{\sigma^2}{\beta^2}.$$

Taking a summation of the above inequality from $t = 1$ to $T - 1$ gives

$$(T-1)T\mathbb{E}[\|\mathbf{w}_T - \mathbf{w}_T^{(S)}\|_2^2] \leq \sigma^2(T-1)/\beta^2.$$

The proof is complete. $\qquad \square$

*Proof of Theorem 6.* It was shown that functions satisfying restricted secant inequality with parameter $\beta$ also satisfies the PL condition with parameter $\beta/L$ (Karimi et al., 2016). Therefore (B.15) holds with $\beta$ there replaced by $\beta/L$. According to Lemma D.2, we know $\mathbb{E}[\|\mathbf{w}_T - \mathbf{w}_T^{(S)}\|_2^2] \leq \sigma^2/(\beta^2 T)$. We can plug this inequality back into (B.15) with $A(S) = \mathbf{w}_{T+1}$, and get

$$\mathbb{E}[F(\mathbf{w}_{T+1})] - F(\mathbf{w}^*) = O\Big(\frac{F(\mathbf{w}^*)}{n\beta} + \frac{1}{\beta^2 T}\Big),$$

where we have used (D.3). By taking $T \asymp n/\beta$, we get $\mathbb{E}[F(\mathbf{w}_{T+1})] - F(\mathbf{w}^*) = O(1/(n\beta))$. This corresponds to $O(n/\beta)$ stochastic gradient evaluations. The proof is complete. $\qquad \square$

*Proof of Theorem 7.* Let $\eta = \beta/L^2$. According to the assumption $\mathbb{E}[\hat{F}_S] = 0$ and (D.1), we know
$$\mathbb{E}[F_S(\mathbf{w}_{t+1})] \leq \mathbb{E}[F_S(\mathbf{w}_t)] - 2\eta\beta\mathbb{E}[F_S(\mathbf{w}_t)] + \eta\beta\mathbb{E}[F_S(\mathbf{w}_t)] = (1 - \eta\beta)\mathbb{E}[F_S(\mathbf{w}_t)]. \quad \text{(D.4)}$$
Applying this inequality recursively, we get $\mathbb{E}[F_S(\mathbf{w}_{T+1})] \leq (1 - \eta\beta)^T \mathbb{E}[F_S(\mathbf{w}_1)]$. We can plug the above inequality back into (3.2) with $A(S) = \mathbf{w}_{T+1}$ and get
$$\mathbb{E}[F(\mathbf{w}_{T+1})] \leq \frac{L\mathbb{E}[F_S(\mathbf{w}_{T+1})]}{2\beta} \leq \frac{L(1 - \beta^2/L^2)^T}{2\beta}\mathbb{E}[F_S(\mathbf{w}_1)] \leq \frac{L\exp(-\beta^2 T/L^2)}{2\beta}\mathbb{E}[F_S(\mathbf{w}_1)],$$
where we have used the elementary inequality
$$1 - a \leq \exp(-a). \quad \text{(D.5)}$$
To achieve $\mathbb{E}[F(\mathbf{w}_{T+1})] \leq \epsilon$, we can take $T$ such that
$$\exp\left(-\beta^2 T/L^2\right) \asymp \epsilon\beta \quad \Longleftrightarrow \quad T \asymp \beta^{-2}\log(1/(\beta\epsilon)).$$
The proof is complete. $\qquad\square$

## D.2 RANDOMIZED COORDINATE DESCENT

We prove here the generalization bounds for randomized coordinate descent. We further assume that the gradient is coordinate-wise Lipschitz continuous in the sense that
$$F_S(\mathbf{w} + \alpha\mathbf{e}_i) \leq F_S(\mathbf{w}) + \alpha\nabla_i F_S(\mathbf{w}) + L\alpha^2/2, \quad \forall\alpha \in \mathbb{R}, \mathbf{w} \in \mathbb{R}^d, i \in [d].$$

*Proof of Theorem 8.* According to Theorem 3 in Karimi et al. (2016), we know
$$\mathbb{E}\big[F_S(\mathbf{w}_{T+1}) - \hat{F}_S\big] \leq \left(1 - \frac{\beta}{dL}\right)^T \mathbb{E}\big[F_S(\mathbf{w}_1) - \hat{F}_S\big]. \quad \text{(D.6)}$$
Plugging the above inequality back into (3.2) and using (D.3), we get
$$\mathbb{E}[F(\mathbf{w}_{T+1})] - F(\mathbf{w}^*) \leq \frac{16L\mathbb{E}\big[\hat{F}_S\big]}{n\beta} + \frac{L}{2\beta}\left(1 - \frac{\beta}{dL}\right)^T \mathbb{E}[F_S(\mathbf{w}_1)]$$
$$= O\left(\frac{F(\mathbf{w}^*)}{n\beta}\right) + O\left(\frac{1}{\beta}\exp\left(-\frac{\beta T}{dL}\right)\right),$$
where we have used (D.5). To achieve the excess generalization bounds $O(1/(n\beta))$, we require $T$ satisfying
$$\exp\left(-\frac{\beta T}{dL}\right) \asymp n^{-1} \quad \Longleftrightarrow \quad T \asymp \frac{d\log n}{\beta}.$$
If $\mathbb{E}[\hat{F}_S] = 0$, then it follows from (3.2), (D.6) and (D.5) that
$$\mathbb{E}[F(\mathbf{w}_{T+1})] \leq \frac{L}{2\beta}\exp\left(-\frac{T\beta}{dL}\right)\mathbb{E}[F_S(\mathbf{w}_1)].$$
To achieve the generalization bound $\epsilon$, we require $T$ satisfying
$$\exp\left(-\frac{T\beta}{dL}\right) \asymp \beta\epsilon \quad \Longleftrightarrow \quad T \asymp \beta^{-1}d\log 1/(\beta\epsilon).$$
The proof is complete. $\qquad\square$

## D.3 STOCHASTIC VARIANCE-REDUCED OPTIMIZATION

We prove here generalization bounds for various stochastic variance-reduced optimization algorithms. We formulate the framework in Algorithm 1.

---
**Algorithm 1:** Stochastic Variance Reduced Optimization

**Input:** step size $\eta$, initialization $\tilde{\mathbf{w}}_0$, $\{m_s\}$

1 **for** $s = 1, 2, \ldots$ **do**
2 $\quad$ set $\mathbf{w}_0 = \tilde{\mathbf{w}}_{s-1}$
3 $\quad$ draw a batch $\tilde{I}_s \subseteq [n]$
4 $\quad$ compute $\mathbf{v}_0 = \nabla f_{\tilde{I}_s}(\mathbf{w}_0)$
5 $\quad$ update $\mathbf{w}_1 = \mathbf{w}_0 - \eta\mathbf{v}_0$
6 $\quad$ **for** $t = 1, \ldots, m_s - 1$ **do**
7 $\quad\quad$ draw a batch $I_t \subseteq [n]$
8 $\quad\quad$ compute $\mathbf{v}_t$ by either (4.4) or (4.5)
9 $\quad\quad$ update $\mathbf{w}_{t+1} = \mathbf{w}_t - \eta\mathbf{v}_t$
10 $\quad$ set $\tilde{\mathbf{w}}_s$ as $\mathbf{w}_{i_s}$, where $i_s$ is drawn according to a distribution on $[m_s]$
11 choose the output from $\{\tilde{\mathbf{w}}_s\}$ according to some strategy

---

We now consider the stochastic variance-reduced gradient descent (SVRG) (Reddi et al., 2016) and stochastically controlled stochastic gradient (SCSG) (Lei et al., 2017).

**Theorem D.3.** *Let Assumptions 1 and 2 hold with $L \leq n\beta/4$. Let $A$ be either the SVRG in Reddi et al. (2016) or the SCSG in Lei et al. (2017). Then we can take $O\big((n + n^{\frac{2}{3}}/\beta) \log n\big)$ stochastic gradient evaluations to get excess generalization bounds $O(1/(n\beta))$. Furthermore, if $\mathbb{E}[\hat{F}_S] = 0$, then we can take $O\big((n + n^{\frac{2}{3}}/\beta) \log 1/(\beta\epsilon)\big)$ stochastic gradient evaluations to achieve the generalization bound $O(\epsilon)$ for any $\epsilon > 0$.*

*Proof.* To achieve $\mathbb{E}[F_S(A(S)) - \hat{F}_S] \leq 2/n$, it was shown that SVRG and SCSG requires $O\big((n + n^{\frac{2}{3}}/\beta) \log n\big)$ stochastic gradient evaluations (Reddi et al., 2016; Lei et al., 2017). We plug this optimization error bound into Theorem 1 and get $\mathbb{E}[F(A(S))] - F(\mathbf{w}^*) = O(1/(n\beta))$.

We now consider the case $\mathbb{E}[\hat{F}_S] = 0$. According to (3.2), to achieve generalization bound $O(\epsilon)$, it suffices that $\mathbb{E}[F(A(S)) - \hat{F}_S] = O(\beta\epsilon)$. This can be achieved by taking $O\big((n + n^{\frac{2}{3}}/\beta) \log 1/(\beta\epsilon)\big)$ stochastic gradient evaluations (Reddi et al., 2016; Lei et al., 2017). The proof is complete. □

We now present the proof of Theorem 9 on the behavior of the stochastic recursive gradient algorithm (SARAH) (Nguyen et al., 2017) and SpiderBoost (Wang et al., 2019).

*Proof of Theorem 9.* To achieve $\mathbb{E}[F_S(A(S)) - \hat{F}_S] \leq 2/n$, it was shown that SARAH and SpiderBoost requires $O\big((n + 1/\beta^2) \log n\big)$ stochastic gradient evaluations (Nguyen et al., 2017; Wang et al., 2019). We plug this optimization error bound into Theorem 1 and get $\mathbb{E}[F(A(S))] - F(\mathbf{w}^*) = O(1/(n\beta))$.

We now consider the case $\mathbb{E}[\hat{F}_S] = 0$. According to (3.2), to achieve generalization bound $O(\epsilon)$, it suffices that $\mathbb{E}[F(A(S)) - \hat{F}_S] = O(\beta\epsilon)$. This can be achieved by taking $O\big((n + 1/\beta^2) \log 1/(\beta\epsilon)\big)$ stochastic gradient evaluations (Nguyen et al., 2017; Wang et al., 2019). The proof is complete. □

Finally, we consider SNVRG-PL (Zhou et al., 2018a).

**Theorem D.4.** *Let Assumptions 1 and 2 hold with $L \leq n\beta/4$. Let $A$ be the SNVRG-PL in Zhou et al. (2018a). Then we can take $O\big((n + \sqrt{n}/\beta) \log^4 n\big)$ stochastic gradient evaluations to get excess generalization bounds $O(1/(n\beta))$. Furthermore, if $\mathbb{E}[\hat{F}_S] = 0$, then we can take $O\big((n + \sqrt{n}/\beta) \log^4 \frac{1}{\beta\epsilon}\big)$ stochastic gradient evaluations to achieve the generalization bound $O(\epsilon)$ for any $\epsilon > 0$.*

*Proof.* To achieve $\mathbb{E}[F_S(A(S)) - \hat{F}_S] \leq 2/n$, it was shown that SNVRG-PL requires $O\big((n + \sqrt{n}/\beta) \log^4 n\big)$ stochastic gradient evaluations (Zhou et al., 2018a). We plug this optimization error bound into Theorem 1 and get $\mathbb{E}[F(A(S))] - F(\mathbf{w}^*) = O(1/(n\beta))$.

We now consider the case $\mathbb{E}[\hat{F}_S] = 0$. According to (3.2), to achieve generalization bound $O(\epsilon)$, it suffices that $\mathbb{E}[F(A(S)) - \hat{F}_S] = O(\beta\epsilon)$. This can be achieved by taking $O\big((n + \sqrt{n}/\beta) \log^4 1/(\beta\epsilon)\big)$ stochastic gradient evaluations (Zhou et al., 2018a). The proof is complete. □

# E  DISCUSSIONS OF EXAMPLES

In this section, we present some discussions on understanding the assumption $L/\beta < n/2$ in Theorem 2.

## E.1  DISCUSSION OF EXAMPLE 1

We first give the definition of strong convexity. For any differentiable function $g : \mathcal{W} \mapsto \mathbb{R}$, we say $g$ is $\sigma$-strongly convex if for any $\mathbf{w}, \mathbf{w}' \in \mathcal{W}$ there holds

$$g(\mathbf{w}') \geq g(\mathbf{w}) + \langle \mathbf{w}' - \mathbf{w}, \nabla g(\mathbf{w}) \rangle + \frac{\sigma}{2} \|\mathbf{w} - \mathbf{w}'\|_2^2.$$

Introduce $g : \mathbb{R}^n \mapsto \mathbb{R}_+$ by $g(\mathbf{v}) = \frac{1}{n} \sum_{i=1}^{n} \ell(v_i, y_i)$. Then the function $F_S$ can be written as

$$F_S(\mathbf{w}) = \frac{1}{n} \sum_{i=1}^{n} \ell(\langle \mathbf{w}, \phi(x_i) \rangle, y_i) = g(A\mathbf{w}),$$

where $A = \left( \phi(x_1), \ldots, \phi(x_n) \right)^\top \in \mathbb{R}^{n \times m}$ is the matrix formed from the data. It is known that if $g$ is $\sigma_g$-strongly convex, then $F_S$ satisfies the PL condition (Karimi et al., 2016; Necoara et al., 2018)

$$F_S(\mathbf{w}) - \hat{F}_S \leq \frac{1}{2\sigma_g \left( \sigma'_{\min}(A) \right)^2} \|\nabla F_S(\mathbf{w})\|_2^2. \tag{E.1}$$

Since $\ell$ is $\sigma_\ell$-strongly convex we know for any $\mathbf{v}, \mathbf{v}' \in \mathbb{R}^n$

$$g(\mathbf{v}') = \frac{1}{n} \sum_{i=1}^{n} \ell(v'_i, y_i) \geq \frac{1}{n} \sum_{i=1}^{n} \ell(v_i, y_i) + \frac{1}{n} \sum_{i=1}^{n} \ell'(v_i, y_i)(v'_i - v_i) + \frac{\sigma_\ell}{2n} \sum_{i=1}^{n} (v_i - v'_i)^2$$

$$= g(\mathbf{v}) + \langle \nabla g(\mathbf{v}), \mathbf{v}' - \mathbf{v} \rangle + \frac{\sigma_\ell}{2n} \|\mathbf{v}' - \mathbf{v}\|_2^2. \tag{E.2}$$

That is, $g$ is $\frac{\sigma_\ell}{n}$-strongly convex. This together with (E.1) shows that

$$F_S(\mathbf{w}) - \hat{F}_S \leq \frac{n}{2\sigma_\ell \left( \sigma'_{\min}(A) \right)^2} \|\nabla F_S(\mathbf{w})\|_2^2 = \frac{1}{2\sigma_\ell \sigma'_{\min}(\Sigma_S)} \|\nabla F_S(\mathbf{w})\|_2^2, \tag{E.3}$$

where we have used

$$\frac{1}{n} \left( \sigma'_{\min}(A) \right)^2 = \frac{1}{n} \sigma'_{\min}(A^\top A) = \sigma'_{\min}(\Sigma_S). \tag{E.4}$$

For any $\mathbf{v}, \mathbf{v}' \in \mathbb{R}^n$, it follows from the $L_\ell$-strong smoothness of $\ell$ that

$$\|\nabla g(\mathbf{v}) - \nabla g(\mathbf{v}')\|_2^2 = \frac{1}{n^2} \sum_{i=1}^{n} \left| \ell'(v_i, y_i) - \ell'(v'_i, y_i) \right|^2 \leq \frac{L_\ell^2}{n^2} \sum_{i=1}^{n} |v_i - v'_i|^2 = \frac{L_\ell^2}{n^2} \|\mathbf{v} - \mathbf{v}'\|_2^2.$$

That is, $g$ is $\frac{L_\ell}{n}$-smooth. It then follows

$$\|\nabla F_S(\mathbf{w}) - \nabla F_S(\mathbf{w}')\|_2 = \left\| A^\top \nabla g(A\mathbf{w}) - A^\top \nabla g(A\mathbf{w}') \right\|_2 \leq \sigma_{\max}(A) \|\nabla g(A\mathbf{w}) - \nabla g(A\mathbf{w}')\|_2$$

$$\leq \frac{L_\ell \sigma_{\max}(A)}{n} \|A(\mathbf{w} - \mathbf{w}')\|_2 \leq \frac{L_\ell \sigma_{\max}^2(A)}{n} \|\mathbf{w} - \mathbf{w}'\|_2.$$

This together with (E.4) shows ($\sigma'_{\min}$ replaced by $\sigma_{\max}$)

$$\|\nabla F_S(\mathbf{w}) - \nabla F_S(\mathbf{w}')\|_2 \leq L_\ell \sigma_{\max}(\Sigma_S) \|\mathbf{w} - \mathbf{w}'\|_2. \tag{E.5}$$

It is reasonable to assume that $L$ is of the order of the smoothness of $F_S$. In this case, it follows from (E.3) and (E.5) that empirical counterpart of $L/\beta$ is of the order of $\sigma_{\max}(\Sigma_S)/\sigma'_{\min}(\Sigma_S)$.

## E.2 DISCUSSION OF EXAMPLE 2

We recall some notations in Example 2. Consider single-hidden-layer neural networks with $d$ inputs, $m$ hidden neurons and a single output, for which the prediction function takes the form $h_{\mathbf{v},\mathbf{w}} = \sum_{k=1}^{m} v_k \phi \left( \langle \mathbf{w}_k, x \rangle \right)$. Here $\mathbf{w}_k \in \mathbb{R}^d$ and $v_k \in \mathbb{R}$ denote the weight of the edges connecting the $k$-th hidden node to the input and output node, respectively, while $\phi : \mathbb{R} \mapsto \mathbb{R}$ is the activation function. We fix $\mathbf{v}$ with $|v_k| = a$ for some $a > 0$ and train $\mathbf{w} = (\mathbf{w}_1, \mathbf{w}_2, \ldots, \mathbf{w}_m)^\top \in \mathbb{R}^{m \times d}$ from $S$. Note we only use the PL condition $F_S(\mathbf{w}) - \hat{F}_S \leq \frac{1}{2\beta} \|\nabla F_S(\mathbf{w})\|_2^2$ for $\mathbf{w} = \mathbf{w}_{S^{(i)}}^{(S^{(i)})}$ in the proof of Theorem 1 (only in (B.6)). We fix $\mathbf{w} = \mathbf{w}_{S^{(i)}}^{(S^{(i)})}$ here. Analogous to Soltanolkotabi et al. (2019), we define the Jacobian matrix $J = \left( J_1, J_2, \ldots, J_n \right) \in \mathbb{R}^{md \times n}$ at $\mathbf{w} = \mathbf{w}_{S^{(i)}}^{(S^{(i)})}$ with

$$J_j = \begin{pmatrix} v_1 \phi'(\mathbf{w}_1^\top x_j) x_j \\ \vdots \\ v_m \phi'(\mathbf{w}_m^\top x_j) x_j \end{pmatrix}$$

and $r_j = \mathbf{v}^\top \phi(\mathbf{w} x_j) - y_j$ for $j \in [n]$. It was shown that (Soltanolkotabi et al., 2019)

$$\nabla F_S(\mathbf{w}) = \frac{1}{n} J\mathbf{r}, \quad \text{for} \quad \mathbf{r} = (r_1, \ldots, r_n)^\top, \tag{E.6}$$

and therefore

$$\|\nabla F_S(\mathbf{w})\|_2^2 = \frac{1}{n^2} \mathbf{r}^\top J^\top J\mathbf{r} = \frac{1}{n^2} \mathbf{r}^\top \left( J_j^\top J_{j'} \right)_{j,j' \in [n]} \mathbf{r}.$$

According to the definition $J_j$, we know

$$J_j^\top J_{j'} = \left( v_1 \phi'(\mathbf{w}_1^\top x_j) x_j^\top, \ldots, v_m \phi'(\mathbf{w}_m^\top x_j) x_j^\top \right) \begin{pmatrix} v_1 \phi'(\mathbf{w}_1^\top x_{j'}) x_{j'} \\ \vdots \\ v_m \phi'(\mathbf{w}_m^\top x_{j'}) x_{j'} \end{pmatrix}$$

$$= a^2 \sum_{k=1}^m \phi'(\mathbf{w}_k^\top x_j) \phi'(\mathbf{w}_k^\top x_{j'}) x_j^\top x_{j'}.$$

It then follows that

$$J^\top J = a^2 \sum_{k=1}^m \left( \phi'(X\mathbf{w}_k)\left(\phi'(X\mathbf{w}_k)\right)^\top \right) \odot (XX^\top), \tag{E.7}$$

where $X = (x_1, \ldots, x_n)^\top \in \mathbb{R}^{n \times d}$ is the data matrix and $\odot$ denotes the Hadamard (entry-wise) product of matrices. According to the definition of $\mathbf{r}$, we know $F_S(\mathbf{w}) = \frac{1}{n}\|\mathbf{r}\|_2^2$. Then, it follows from (E.6) and (E.7) that

$$\|\nabla F_S(\mathbf{w})\|_2^2 = \frac{a^2}{n^2} \mathbf{r}^\top \left( \sum_{k=1}^m \left( \phi'(X\mathbf{w}_k)\left(\phi'(X\mathbf{w}_k)\right)^\top \right) \odot (XX^\top) \right) \mathbf{r}$$

$$\geq \frac{a^2}{n^2} \sigma_{\min} \left( \sum_{k=1}^m \left( \phi'(X\mathbf{w}_k)\left(\phi'(X\mathbf{w}_k)\right)^\top \right) \odot (XX^\top) \right) \|\mathbf{r}\|_2^2$$

$$= \frac{a^2}{n} \sigma_{\min} \left( \sum_{k=1}^m \left( \phi'(X\mathbf{w}_k)\left(\phi'(X\mathbf{w}_k)\right)^\top \right) \odot (XX^\top) \right) F_S(\mathbf{w}).$$

That is, we can take the parameter of the PL condition as

$$\beta = \frac{a^2}{2n} \sigma_{\min} \left( \sum_{k=1}^m \left( \phi'(X\mathbf{w}_k)\left(\phi'(X\mathbf{w}_k)\right)^\top \right) \odot (XX^\top) \right).$$

It is reasonable to assume that $L$ is of the order of $\frac{a^2}{n}\sigma_{\max}\left( \sum_{k=1}^m \left( \phi'(X\mathbf{w}_k)\left(\phi'(X\mathbf{w}_k)\right)^\top \right) \odot (XX^\top) \right)$ (Soltanolkotabi et al., 2019). In this case, we have the empirical counterpart of $L/\beta$ is of the order of

$$\frac{\sigma_{\max}\left( \sum_{k=1}^m \left( \phi'(X\mathbf{w}_k)\left(\phi'(X\mathbf{w}_k)\right)^\top \right) \odot (XX^\top) \right)}{\sigma_{\min}\left( \sum_{k=1}^m \left( \phi'(X\mathbf{w}_k)\left(\phi'(X\mathbf{w}_k)\right)^\top \right) \odot (XX^\top) \right)}.$$

If we consider the identify activation function, i.e., $\phi(t) = t$, then it follows from the definition of $\Sigma_S$ that

$$L/\beta \asymp \frac{\sigma_{\max}(XX^\top)}{\sigma_{\min}(XX^\top)} = \frac{\sigma_{\max}(\Sigma_S)}{\sigma_{\min}(\Sigma_S)}.$$

