# OpenReview forum: "Sharper Generalization Bounds for Learning with Gradient-dominated Objective Functions"
_ICLR.cc/2021/Conference — ICLR 2021 Poster_

### Official Review · AnonReviewer2 · 2020-10-26
**Good theory on the study of generalization error, but experiments are missing**

**Rating:** 5
**Confidence:** 3

**Review:**



The whole paper is focused on the theoretical part. I am missing the important experiment part to back up the theory. I could not vote for acceptance without the experiments, and I will increase the score if satisfying experiment results can be included.

It is also important to verify the condition $L \leq n\beta/4$ in the experiment, i.e. how large the sample size needs to be. In terms of this constraint, I saw that the authors tried to investigate and interpret it. But I still have some questions on them. Please correct me if any of my understanding is wrong:

Both $L$ and $\beta$ seem to depend on $n$ and are not absolute constants here. For example $L/\beta$ is about $\sigma_{\max}(\Sigma_S) / \sigma_{\min}(\Sigma_S)$ in the examples. Here my understanding is that $L/\beta$ could be large if $n$ is small. When $n$ becomes super large then $L/\beta$ would be close to some constant

- if n is super large, then the constraint $L \leq n\beta/4$ would be satisfied. But since n is super large, this constraint is also very trivial and seems not imposing any "implicit constraint" on the complexity of the problem.
- if n is moderate, then this constraint $L \leq n\beta/4$ might not be satisfied

---

> ### Author Response · Authors · 2020-11-16
> **Response to Reviewer 2**
>
> Thank you for your invaluable comments and suggestions.
>
>
> **Q: I am missing the important experiment part to back up the theory. It is also important to verify the condition $L\leq n\beta/4$.** \
> **A:** We agree it is important to verify the condition $L/\beta\leq n/4$. Theoretically, the empirical covariance matrix $\Sigma_S$ would converge to the covariance matrix $\Sigma=\mathbb{E}[xx^\top]$ as $n$ increases, which is independent of $n$. Therefore, we would expect that $\sigma_{\text{max}}(\Sigma_S)/\sigma_{\text{min}}(\Sigma_S)$ would be close to some constant for large $n$. The constraint $L/\beta\leq n/4$ would then be easy to satisfy for large $n$, and may violate if $n$ is small or moderate.
>
> We have added two experiments to support our theory. In the first experiment, we aim to verify how the assumption $\sigma_{\text{max}}(\Sigma_S)/\sigma_{\text{min}}(\Sigma_S)\leq n/4$ would hold in practice. Experimental results show that this assumption may be violated if $n$ is small or moderate, and hold trivially for sufficiently large $n$.
>
> In the second experiment, we aim to verify Theorem 1 on the resistance of stochastic optimization to overfitting under the PL condition. We consider generalized linear models, which are nonconvex and are shown to satisfy the PL condition (Foster et al., 2019). Our experimental results show that there is no overfitting even if we run SGD with $100$ passes, which is consistent with our bound in Theorem 1.
>
> Please see Section 5 for details.

---

### Official Review · AnonReviewer3 · 2020-10-28
**Review for 'Sharper Generalization Bounds for Learning with Gradient-dominated Objective Functions'**

**Rating:** 6
**Confidence:** 3

**Review:**

This paper studies the generalization performance of stochastic algorithms in nonconvex optimization with gradient dominance condition. In detail, the authors suggest that for any algorithm, its generalization error can be bounded by $O(1/(n\beta))$ plus the optimization error of the algorithm, where $\beta$ is the gradient dominance parameter. The main idea for the authors to obtain such an improved bound is an advanced analysis based on a weaker on-average stability measure.

Here are my detailed comments.
- It seems that to do a comparison, the authors need to ‘translate’ previous complexity results such as stability results (Charles & Papailiopoulos, 2018) to generalization results, considered in this paper. To do such a translation, Theorem A.1 is proposed. However, I did not find the proof to Theorem A.1 except its part c. Since the translation is crucial for a fair comparison, the authors may want to complete the proof to Theorem A.1 or point out some relevant references about that.
- Assumption 1 and 2 suggest that $L$ and $\beta$ are two parameters about $F$ itself, in other words, independent of any specific data set $S$. However, in Example 1 and 2, the authors suggest that $L/\beta$ is in the order of $\sigma_{max}(\Sigma_S)/\sigma_{min}(\Sigma_S)$, which depends on the random data $S$. Can the authors explain more about that? Shall we need some more assumptions about the data distribution (for instance, the support set of $x$ is finite) to support such a claim?
- $O(1/(n\beta))$ generalization gap is standard for the case where $F$ is convex, and this work tries to extend it from convex to nonconvex with PL condition, just as the authors suggest in Remark 3. Does the $O(1/(n\beta))$ gap still hold under other nonconvex relaxing conditions such as one-point strongly convexity or quadratic growth [1]?

[1] Necoara, Ion, Yu Nesterov, and Francois Glineur. "Linear convergence of first order methods for non-strongly convex optimization." Mathematical Programming 175.1-2 (2019): 69-107.

---

> ### Author Response · Authors · 2020-11-16
> **Response to Reviewer 3**
>
> Thank you for the comments and constructive suggestions.
>
> **Q: The authors may need to "translate" previous stability results to generalization results. The authors may want to complete the proof to Theorem A.1 or point out some relevant references about that.** \
> **A:** Yes, you are right. We need Theorem A.1 to translate previous bounds in the comparison. We agree it is important to either prove or give the exact reference to these crucial results. The proof of Part (a) can be found in Hardt et al., (2016, Theorem 2.2). Part (b) was first proved for deterministic algorithms (Bousquet \& Elisseeff 2002, Theorem 11), and then extended to randomized algorithms (Elisseeff et al. 2005, Theorem 12). We have added these references in the proof of Theorem A.1.
>
> **Q: $L,\beta$ in Assumptions 1, 2 are parameters about $F$, while $\Sigma_S$ depends on random $S$ in Examples 1, 2.  Shall we need some more assumptions about the data distribution?** \
> **A:** Sorry for the confusion. The parameter $L$ reflects the smoothness of individual loss functions, while $\sigma_{\text{max}}(\Sigma_S)$ is related to the smoothness of $F_S$. In our discussion of examples, we assume $L$ and $\sigma_{\text{max}}(\Sigma_S)$ are of the same order (Appendix E). Note $L$ can be small in practice, e.g., least square loss and logistic loss. For the PL condition, the parameter $\beta$ is more related to $F_S$ instead of $F$ since Assumption 2 involves the term $\inf_{\mathbf{w}}F_S(\mathbf{w})$ and $\|\nabla F_S(\mathbf{w})\|_2^2$. We take an expectation w.r.t. $S$ in Assumption 2 since we only study generalization bounds in expectation. In Examples 1, 2, we show that the gradient-dominance parameter of $F_S$ is related to $\sigma_{\text{min}}(\Sigma_S)$. We have modified these examples by replacing "$L/\beta$ is of the order of" with "the empirical counterpart of $L/\beta$ is of the order of" to avoid the confusion.
>
> Our analysis does not require more assumptions on the data distribution. Indeed, our bounds are distribution-free.
>
> **Q: $O(1/(n\beta))$ is standard if $F$ is convex. Does the gap still hold under other nonconvex relaxing conditions such as one-point strongly convexity or quadratic growth?** \
> **A:** Firstly, we would like to clarify that convexity is not sufficient to guarantee the generalization gap $O(1/(n\beta))$. An additional assumption like strong convexity is needed for the $O(1/(n\beta))$ generalization gap.
>
> The $O(1/(n\beta))$ gap also holds under the one-point strong convexity assumption. Indeed, one-point strong convexity and smoothness imply the PL condition (Yuan et al., 2019, Lemma 2). Therefore, all our results apply to one-point strongly convex functions.
>
> For functions satisfying the quadratic growth condition (Necoara et al., 2019), we need another realizability assumption to enjoy generalization. We have added new generalization bounds in Theorem 3 on the quadratic growth condition. Note that the realizability condition is also required in Charles \& Papailiopoulos (2018) to study the generalization gap for functions satisfying the quadratic growth condition.

---

### Official Review · AnonReviewer1 · 2020-10-28
**Impressive Results and Well-Writen Paper**

**Rating:** 7
**Confidence:** 3

**Review:**

This paper mainly studies the generalization performance of stochastic algorithms. Compared with previous results which rely on Lipschitz condition, this paper assumes smoothness condition and Polyak-Lojasiewicz Condition, and then prove the excess generalization bound that is a summation of $\frac{1}{n\beta}$ and empirical optimization error. This result looks impressive, not only because the first term looks shaper than previous $\frac{1}{\sqrt{n}}$ of generalization bound , but also it implies optimization benefits generalization, which may help understand some empirical observations in modern machine learning. What's more, authors analyze some common stochastic algorithms as concrete examples to show the corresponding theoretical guarantee.  Besides, the whole paper is well-written and easy to follow.

Though results in this paper look stronger than previous results, authors use different assumptions (say smoothness instead of Lipschitz condition) compared with previous paper, and it would be better if authors could make these comparisons more clear (say in abstract etc.). Besides, though there is a $\frac{1}{n\beta}$ term in the generalization bound, it holds in expecation, and I think an extension to high probability bound will lead to an additional $\frac{1}{\sqrt{n}}$ term, which should also be stated clear when comparing with previous results. Finally, I wonder whether it is possible to extend the results to non-smooth loss function, at least for some special case, such as DNN with ReLU activation unit.

---

> ### Author Response · Authors · 2020-11-16
> **Response to Reviewer 1**
>
> Thank you for your constructive comments.
>
> **Q: It would be better to make comparisons more clear by stating clearly the different assumptions on smoothness and Lipschitz continuity.** \
> **A:** We add a sentence in the abstract to clarify the different assumptions. The added sentence is "We achieve this improvement by exploiting the smoothness of loss functions instead of the Lipschitz condition in Charles \& Papailiopoulos (2018)." We also add more discussions in the "generalization analysis" part of Section 2 to make the comparisons more clear.
>
> **Q: An extension to high probability bound will lead to an additional $1/\sqrt{n}$ term.** \
> **A:** We make this clear by adding the following sentences at the end of Remark 1: ``As compared to probabilistic bounds in Charles \& Papailiopoulos (2018), our bounds are stated in expectation. The extension to high-probability bounds will lead to additional $O(1/\sqrt{n})$ term (Feldman \& Vondrak, 2019)''
>
> **Q: Extension to non-smooth setting.** \
> **A:** We extend Theorem 1 to a Lipscthiz and non-smooth setting (Theorem 4). In this setting, we show that generalization errors can be bounded by $O\big(\frac{G^2}{n\beta}+G\sqrt{\frac{\epsilon_{\text{opt}}}{\beta}}\big)$, where $G$ is the Lipschitz constant and $\epsilon_{\text{opt}}$ is the optimization error. This differs from Theorem 1 in two aspects. First, the minimal training error in Theorem 1 is replaced by $G^2$. Second, the term $\frac{\epsilon_{\text{opt}}}{{\beta}}$ in Theorem 1 is replaced by $\sqrt{\frac{\epsilon_{\text{opt}}}{{\beta}}}$. Note Theorem 4 is also able to imply limiting generalization bounds $O(1/(n\beta))$ if $\epsilon_{\text{opt}}$ is sufficiently small, which outperforms the bound $O(1/\sqrt{n\beta})$ in Charles \& Papailiopoulos (2018). Please see Theorem 4 for details.

---

### Official Review · AnonReviewer4 · 2020-11-05
**Provide new understanding on generalization, however need more experiments**

**Rating:** 6
**Confidence:** 2

**Review:**

One of the main contributions of this paper lies in the generalization error of O(1/n) under the P-L condition given by Theorem 1. It provides a generalization bound instead of the hypothesis stability or uniform stability in previous discussions. The result explains generalization even under overparameterized model.


Strength:
+ The theoretical result is novel, according to the authors this is the first paper to model the generalization bound in terms of a $O(1/n\beta)$ term plus the convergence rate of the optimization algorithm
+ The authors applies the main theorem on practical algorithms and derive convergence rate on generalization error while previous literature on those algorithms are mainly on  training error.


Weakness:
- Some of the background description and preliminaries are vague, for example the $E_{S, A}$ concerning taking an expectation over an algorithm A are not explained.
- The relation of related works with this work is not clear enough. The two paragraphs, "Algorithmic Stability" and "Generalization Analysis" all ends up with areas that stability/generalization is applicable, while I care more about how previous literature shed lights on this work.
- I see no experimental results to support their theory

I think it is a good paper that provides novel aspects of understanding. It provides new insights in understanding important phenomenons in training deep neural networks whose Lipchitz constant might be very large. The bound removes the Lipchitz constant and uses training error of the best model instead. It explains why interpolation does not lead to overfitting. However the first part of this paper is a little bit rash while the experimental support is insufficient.

Question:

How widely is the PL condition applicable? Are there any examples when the PL condition does not hold?

In Theorem 1 the assumption is $L \leq n\beta/4$, while in Corollary 2 the assumption becomes $L < n\beta/2$, what makes the difference?

---

> ### Author Response · Authors · 2020-11-16
> **Response to Reviewer 4**
>
> Thank you for your invaluable and constructive comments.
>
> **Q: Some of the background description and preliminaries are vague, e.g., the expectation w.r.t. the algorithm.** \
> **A:** We have checked the background description and made necessary changes. We have added more explanation on the expectation w.r.t. $A$. Here is the added sentence "Here $\mathbb{E}_A$ denotes the expectation w.r.t. the randomness of the algorithm $A$. For example, if $A$ is SGD, then $\mathbb{E}_A$ denotes the expectation w.r.t. the random indices of training examples selected for the gradient computation." Please see the first paragraph of Section 3.
>
> **Q: In the related work, I care more about how previous literature shed lights on this work.** \
> **A:** Accordingly, we have re-organized the related work on algorithmic stability. In this revision, we group the discussions according to stability measures. We first discuss the related work on uniform stability. Then we move onto several other stability measures. Finally, we emphasize the on-average stability by indicating its advantage in exploiting either the strong convexity or the relaxed exp-concavity in the existing analysis. This motivates us to use the on-average stability for learning with the PL condition, which is another relaxed condition of strong convexity.
>
> We have separated the related work on generalization analysis in two paragraphs. The first paragraph is on the stability approach, and the second paragraph sketches some other approaches. In the end of the first paragraph, we put more discussions on the most related work (Charles \& Papailiopoulos, 2018) by presenting some defects. We explain that these defects are due to the use of pointwise hypothesis stability and the ignorance of smoothness property, which motivate us to refine their results by considering the weaker on-average stability and exploiting the smoothness of loss functions.
>
> Please see Section 2 for details.
>
> **Q: How widely is the PL condition applicable? Are there any examples when the PL condition does not hold?** \
> **A:** PL condition is widely used in the non-convex learning setting (Zhou et al., 2018b; Reddi et al., 2016; Karimi et al., 2016;
> Wang et al., 2019; Lei et al., 2017, Charles \& Papailiopoulos, 2018; Yuan et al., 2019). It is shown to hold for strongly convex functions composed with piecewise-linear objectives (Karimi et al., 2016, Charles \& Papailiopoulos, 2018), and non-convex learning problems including generalized linear models, robust regression (Foster et al., 2019) and linear neural networks (Charles \& Papailiopoulos, 2018, Hardt \& Ma, 2016).
>
> PL condition can fail for very simple functions. For example, consider the one-dimensional function $f(x)=x^4$. Then $f'(x)=4x^3$ and the minimal value is $0$. Furthermore, we have
> $$
> \frac{f(x)-0}{|f'(x)|^2}=\frac{x^4}{16x^6}=\frac{1}{16x^2}\to\infty\text{ as }x\to0.
> $$
> Therefore, the PL condition does not hold.
>
> **Q: The difference between the condition $L\leq n\beta/4$ and $L<n\beta/2$.** \
> **A:** Under the condition $L<n\beta/2$, our analysis in the proof of Theorem 1 can actually imply the bound
> $$
> \mathbb{E}\big[F(\mathbf{w}_S) - \hat{F}_S\big]\leq
> \frac{8L\mathbb{E}[\hat{F}_S]}{n\beta-2L} + \frac{L\mathbb{E}\big[F_S(\mathbf{w}_S)-\hat{F}_S\big]}{2\beta}.
> $$
> This implies the bound in Theorem 1 by taking a further assumption $L\leq n\beta/4$. In Corollary 2, we assume $\mathbb{E}[\hat{F}_S]=0$ and therefore we can get $\frac{8L\mathbb{E}[\hat{F}_S]}{n\beta-2L}=0$ under the condition $L<n\beta/2$. We have added a note below the proof of Theorem 1 to make this clear (the top of Page 17).
>
> **Q: I see no experimental results to support their theory.** \
> **A:** We have added two experiments to support our theory. The first experiment shows that the condition $L/\beta\leq n/4$ can be violated if $n$ is small, and holds if $n$ is large. We observe no overfitting for learning with gradient-dominated functions in the second experiment, which is consistent with Theorem 1 on the resistance of SGD to overfitting under the PL condition. Please see Section 5 for details.

---

### Decision · Program_Chairs · 2021-01-10
**Final Decision**

**Decision:**

Accept (Poster)

**Comment:**

In this work, the authors develop an improved generalization bound for stochastic optimization algorithms. Reviewers agree that the theoretial results are significant. Several reviewers had concerns about the lack of experimental validation, which the authors addressed during the discussion phase. Other more minor concerns were also adequately addressed by the authors. The final recommendation is therefore to accept.